Methods

# Efficient knock-in method enabling lineage tracing in zebrafish

Jiarui Mi, Olov Andersson

Here, we devised a cloning-free 3′ knock-in strategy for zebrafish using PCR amplified dsDNA donors that avoids disrupting the targeted genes. The dsDNA donors carry genetic cassettes coding for fluorescent proteins and Cre recombinase in frame with the endogenous gene but separated from it by self-cleavable peptides. Primers with 5′ AmC6 end-protections generated PCR amplicons with increased integration efficiency that were coinjected with preassembled Cas9/gRNA ribonucleoprotein complexes for early integration. We targeted four genetic loci (*krt92*, *nkx6.1*, *krt4*, and *id2a*) and generated 10 knock-in lines, which function as reporters for the endogenous gene expression. The knocked-in iCre or CreERT2 lines were used for lineage tracing, which suggested that *nkx6.1*[+] cells are multipotent pancreatic progenitors that gradually restrict to the bipotent duct, whereas *id2a*[+] cells are multipotent in both liver and pancreas and gradually restrict to ductal cells. In addition, the hepatic *id2a*[+] duct show progenitor properties upon extreme hepatocyte loss. Thus, we present an efficient and straightforward knock-in technique with widespread use for cellular labelling and lineage tracing.

## Introduction

With the advent of genome editing tools, the CRISPR-Cas9 system has become the most popular method for the generation of genetically engineered animal models (Wang et al, 2013; Yang et al, 2013), where we refer to "knock-in" as the insertion of exogenous DNA at a specific locus in the host genome. Multiple knock-in strategies have been introduced and optimized for the construction of nonhuman primate (Zuo et al, 2017; Yao et al, 2018a), mouse (Zuo et al, 2017), medaka (Gutierrez-Triana et al, 2018; Seleit et al, 2021; Tan & Winkler, 2022), and zebrafish models (Irion et al, 2014; Hoshijima et al, 2016; Wierson et al, 2020; Almeida et al, 2021). In zebrafish, the knock-in methods vary in terms of the targeting regions (such as 5′ noncoding region, exon, intron, or 3′ end) (Han et al, 2021; Levic et al, 2021), DNA double-stranded break repair

mechanisms (homology-directed repair [HDR] or nonhomologous end joining [NHEJ]) (Auer et al, 2014), the type of donor templates (Wierson et al, 2020), the injection of Cas9 protein or Cas9 mRNA, and the use of drugs in promoting HDR (Albadri et al, 2017; Aksoy et al, 2019).

In zebrafish, 5′ knock-in methods have been intensively investigated in the locus upstream of the start codon ATG using donor plasmids containing in vivo linearization site(s) (Auer et al, 2014; Hisano et al, 2015). The single linearization site upstream of the insertion sequence can facilitate NHEJ-mediated integration (Irion et al, 2014; Kimura et al, 2014; Kesavan et al, 2017); researchers also tried to introduce long homologous arms (HAs) flanked by two I-SceI/gRNA recognition sites to induce HDR (Hoshijima et al, 2016). Although fluorescent reporter lines, and even CreERT2 lines, have been generated by such methods (Kesavan et al, 2018), the wide applications of these methods are still hampered by the disruption of one allele of the endogenous gene and the multiple molecular cloning steps. The 3′ knock-in method has also been applied using circular plasmids as the donor, with either long or short HAs flanked by two in vivo linearization sites (Eschstruth et al, 2020; Gillotay et al, 2020). The advantage of 3′ knock-in is that it keeps the knock-in cassettes in-frame and maintains the functionality of the endogenous gene. However, in certain cases, a few amino acids in the C-terminus may be deleted when using the NHEJ strategy (Cronan & Tobin, 2019). Several studies reported that the HDR-mediated 3′ knock-in efficiency is highly dependent on the length of the HAs (usually with higher efficiency using >500 bp HAs) (Irion et al, 2014; Hoshijima et al, 2016). Nevertheless, one recent study showed that the introduction of short HAs in the donor plasmids flanked by two linearization sites can enhance microhomology-mediated end joining (MMEJ) with good efficiency (Luo et al, 2018). However, such methods are still somewhat limited in use because of the low scalability and the complex construct preparation steps. Recently, intron-based and exon-based knock-in approaches have remarkably expanded the knock-in toolbox by targeting genetic loci beyond the 5′ or 3′ end (Li et al, 2015; Li et al, 2019; Welker et al, 2021). These methods mostly rely on the NHEJ method, and the endogenous genes can be either destroyed or rescued (depending on whether the exon sequences downstream of the insertion site are added into the donor or not). Given that all these methods are

---

Department of Cell and Molecular Biology, Karolinska Institutet, Stockholm, Sweden

Correspondence: olov.andersson@ki.se

limited in scalability and involve multiple molecular cloning steps in construct preparation, the development of a straightforward and efficient knock-in methodology is still warranted in the zebrafish field.

Recent studies in mouse and in vitro systems have demonstrated several approaches that improve HDR. The Tild-CRISPR (targeted integration with linearized dsDNA-CRISPR) strategy that used PCR-amplified or enzymatic-cut donors with 800-base pair HAs have been successfully applied in generating knock-in mouse lines (Yao et al, 2018b), indicating that nude double-stranded DNA (dsDNA) can serve as an effective donor in eukaryote embryos. Furthermore, 5′ modified dsDNA with short HAs (roughly 50 base pairs) demonstrated impressive knock-in efficiency in an in vitro culture system (Yu et al, 2020). In this study, the researchers systematically compared 13 modifications to dsDNA with gRNA targeting the 3′ UTR of the GAPDH gene in HCT116 cells. The dsDNAs were synthesized by PCR amplification with the modifications incorporated in the primers. It showed that C6 linker (AmC6) or C12 linker (AmC12) and moieties by adding on secondary modifications outperformed no C6/C12 linked modifications with a substantial increase of knock-in efficiency of more than fivefold. Although the mechanism is still undetermined, it is postulated that the 5′ modification can help prevent degradation and multimerization of the donor and circumvent stochastic NHEJ, indicated by less NHEJ events and random insertions.

Inspired by these previous efforts to improve knock-in efficiency (Yu et al, 2020), here, we introduce a straightforward CRISPR-Cas9-guided 3′ knock-in approach to generate zebrafish lines for cellular labelling and lineage tracing. We synthesized 5′ modified dsDNA with either short or long HAs as the donor by a simple PCR step with 5′ modified primers (AmC6). The donor templates code for two kinds of 2A peptides linking the endogenous gene product with a fluorescent protein and then with iCre/CreERT2. By coinjecting this type of donor with in vitro preassembled Cas9/gRNA ribonucleoprotein complexes (RNPs), we generate mosaic F0 with very high probability of giving rise to germline transmission. Ultimately, we generated 10 knock-in fish lines, demonstrating high scalability. Our knock-in lines can precisely reflect the endogenous gene expression, as visualized by optional fluorescent proteins. Importantly, we also performed lineage-tracing experiments using the knock-in iCre and CreERT2 lines to delineate cell differentiation paths in pancreas and liver development and injury models.

## Results

### A 3′ knock-in pipeline and the characterization of *TgKI(krt92-p2A-EGFP-t2A-CreERT2)*

With the aim to generate knock-in zebrafish lines for both cellular labelling and lineage tracing, we designed our vector templates encompassing a fluorescent protein and different Cre recombinases linked by two self-cleavable 2A peptide sequences (p2A and t2A) (Fig 1A). The insertion sequences were flanked by left and right long HAs (nearly 900 base pairs) on the basis of the GRCz11 reference genome archived in the Ensembl database. Next, we used

pairs of 5′ AmC6 modified primers to amplify dsDNA with the insertion sequence flanked by either long or short HAs by PCR. Subsequently, we coinjected the PCR product, which serves as the direct donor, together with in vitro preassembled Cas9/gRNA RNPs into the one-cell stage zebrafish embryos (of the TL strain). Whereas many of the injected embryos can display some fluorescence/integration, we aim to sort out the F0 with high mosaicism (estimated as >30%) based on the fluorescence in the expected cell types and raise them up. A handful of adult F0 fish were then to be outcrossed with WT fish to screen for founders. To test the validity and efficiency of this pipeline, we first selected an epithelial marker, *krt92*, as it facilitated the identification of mosaic F0 by a simple detection of fluorescence in the skin.

For the *krt92* locus, the dsDNA donor and gRNA sequence are shown in Figs 1B and S1. We selected one gRNA 20 base pairs upstream of the stop codon. To circumvent the cleavage of the donor, and keep the endogenous amino acid sequence intact, we incorporated several synonymous point mutations in the left HA (Fig S1). We used long HAs on both sides to enhance the annealing of the sequences. With 5′ modified dsDNA injection, we observed that 8 out of 158 (5.1%) injected embryos showed fluorescence in approximately a third of the skin, suggesting early integration (Fig 1C). Among them, four fish were identified as founders by screening more than 200 F1 embryos from each fish (Fig 1D). The proportion of the F1 generation that carried the knock-in cassette ranged from 11.5% to 20% (Fig 1E). Sanger sequencing confirmed the correct in-frame integration at the junction between the endogenous gene and the 5′ end of the integrated sequence (Fig S2 and data uploaded to a public repository: https://osf.io/tdkvh/).

To validate the functionality of Cre recombinase, we crossed *TgKI(krt92-p2A-EGFP-t2A-CreERT2)* with the responder line *Tg(ubb: loxP-CFP-STOP-Terminator-loxP-hmgb1-mCherry)* (abbreviated as *Tg(ubi:CSHm)*) (Fig 1F). The cells with *krt92* expression during the 4-hydroxytamoxifen (4-OHT) treatment were expected to have Cre recombinase translocated to the nucleus to conduct recombination. After recombination, all the *krt92*⁺ cell progeny would express H2BmCherry as it is directly driven by the *ubiquitin B* promoter (Mosimann et al, 2011). The H2BmCherry signal was detected in skin cells after 4-OHT administered at 6 hours postfertilization (hpf) (Figs 1G–J and S3). We also observed that various proportions of intestinal cells were fluorescently labelled after 4-OHT treatments at different timepoints (Fig S4). Although in situ hybridization for *krt92* is not feasible because of the high sequence similarity with other keratins, the fluorescence pattern matched the recent single-cell RNA-seq data of the zebrafish intestine, showing a widespread expression across different intestinal epithelial cell types (Wen et al, 2021; Willms et al, 2022). Lastly, we noticed that neither the circular plasmid with in vivo linearization sites (indicated by gRNA1 and gRNA2) nor the dsDNA without 5′ end protection could achieve successful integration (Fig 1B and C). In summary, the dsDNA with 5′ modifications is an efficient donor for generating HDR-dependent knock-in zebrafish lines.

### The generation of *nkx6.1* knock-in lines using short or long HAs

Next, we aimed to knock-in donors at the 3′ end of *nkx6.1*, which is a transcription factor essential in the development of the pancreas

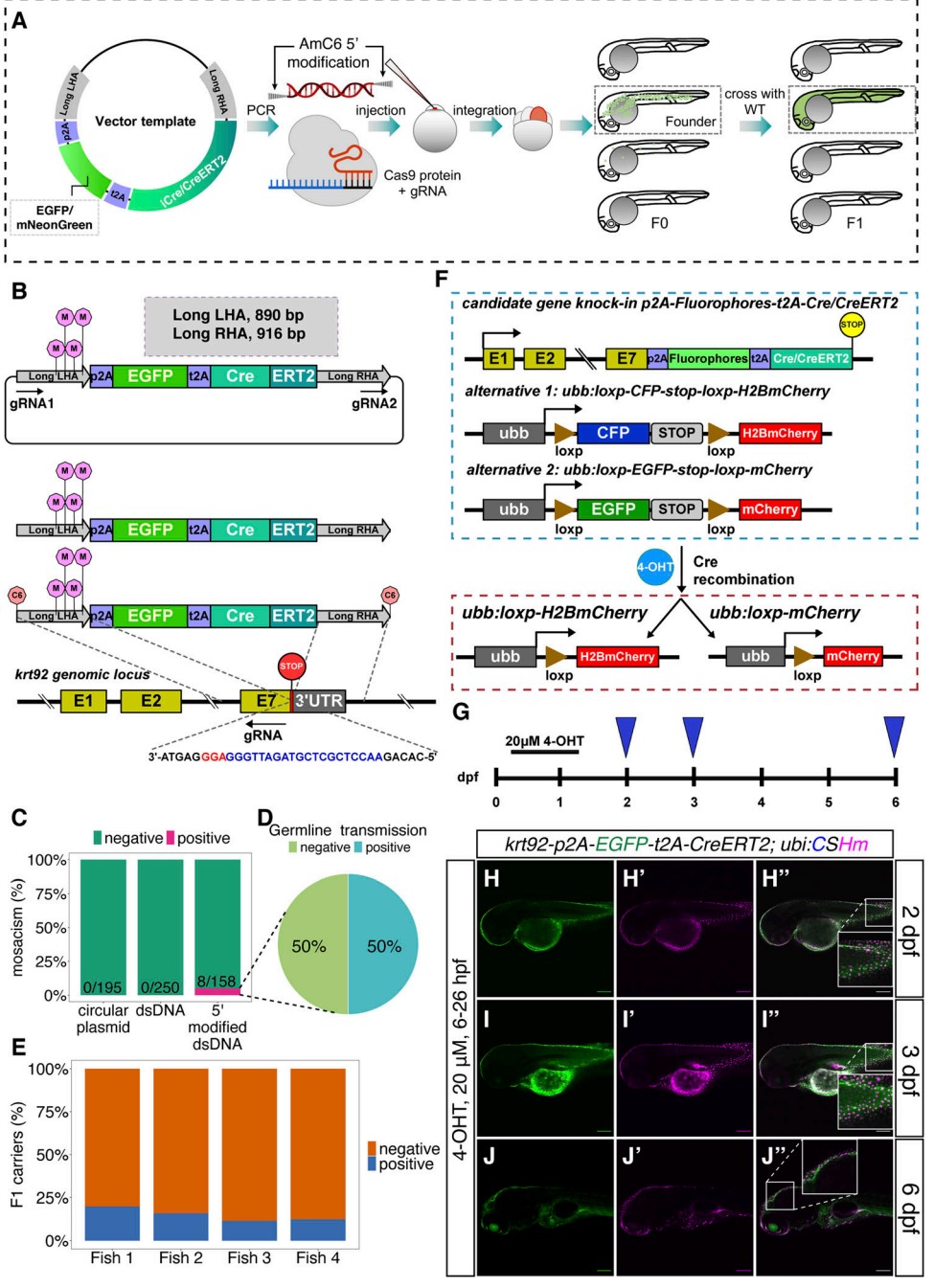

**Figure 1. Knock-in pipeline and characterization of knock-in line at the *krt92* locus.**
**(A)** Schematic representation of the 3′ knock-in pipeline with 5′ modified dsDNA as the donor. The iCre indicates improved Cre.
**(B)** The design of the template vector, PCR-amplified dsDNA with 5′ modifications, and the gRNA sequence for the construction of *TgKI(krt92-p2A-EGFP-t2A-CreERT2)*. The gRNA1 and gRNA2 indicate in vivo linearization sites (sequences listed in Table S2). The pink lollipops in the middle of long LHA indicate synonymous point mutations on the left HAs. The orange lollipops at the end of dsDNAs indicate 5′ AmC6 modifications. The nucleotide sequence in blue indicates gRNA; whereas the nucleotide sequence in red indicates the PAM sequence.
**(C, D, E)** Summary statistics of *krt92* knock-in efficiency using different donors, including the percentage of injected F0 with at least 30% fluorescence labelling in the skin (C), the percentage of adult F0 giving rise to germline transmission (D), and the percentage of F1 siblings from four different founders (Fish 1–4) carrying the knock-in cassette (E). **(F)** The scheme for the lineage-tracing strategy for the iCre or tamoxifen-inducible Cre knock-in lines. There are two alternatives for the color switch using the Cre responder lines, option 1 contains *ubb:loxp-CFP-stop-loxp-H2BmCherry* transgene (abbreviated as *ubi:CSHm*), whereas option two contains *ubb:loxp-EGFP-stop-loxp-mCherry* (abbreviated as *ubi:Switch*). Cells with Cre recombination will ubiquitously express H2BmCherry or mCherry. **(G, H, I, J)** Temporal labelling with 20 μM 4-OHT treatment at 6–26 hpf in *TgKI(krt92-p2A-EGFP-t2A-CreERT2);Tg(ubi:CSHm)* line (G); and representative confocal images at 2 dpf (H–H″), 3 dpf (I–I″), and 6 dpf (J–J″). Skin cells with Cre recombination after 4-OHT treatment were labelled with H2BmCherry. The insets are magnified views showing the expression pattern of two fluorescent proteins. Scale bars = 200 μm.

and motor neurons (Cheesman et al, 2004). We selected a gRNA spanning over the stop codon region and used 5′ modified dsDNA with long HAs to generate *TgKI(nkx6.1-p2A-EGFP-t2A-CreERT2)* (Fig 2A). 2 out of 1,000 (0.2%) *nkx6.1-p2A-EGFP-t2A-CreERT2*-injected F0 embryos using long HAs showed detectable fluorescent signals in at least 30% of the spinal cord (Fig 2B). Overall, it was harder to estimate the percentage mosaicism when the expression was low and present in tissues with cells overlaying each other, although this did not affect the establishment of the line.

Because the donor plasmids with short HAs flanked by in vivo linearization sites were also applicable in zebrafish knock-ins by inducing MMEJ (Nakade et al, 2014; Sakuma et al, 2016; Luo et al, 2018), we then converted to dsDNA with short HAs generated by a simple PCR step with primers containing 41 and 33 base pairs flanking the insertion sequence. We injected the dsDNA donor carrying *p2A-mNeonGreen* and *p2A-mNeonGreen-t2A-iCre* cassettes and, strikingly, we noted a dramatic increase in the frequency of embryos showing mosaic fluorescence reporter expression in the spinal cord, that is, *nkx6.1-p2A-mNeonGreen-t2A-iCre* (in 5 out of 355

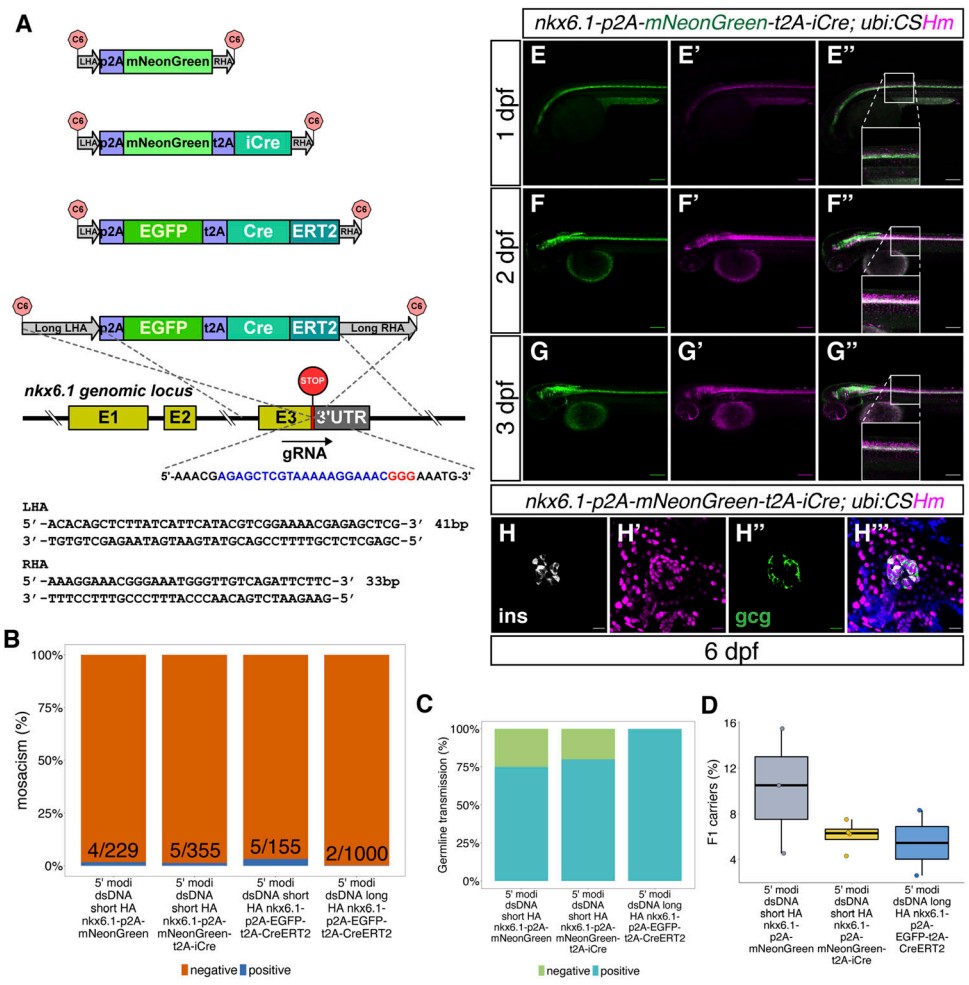

**Figure 2. The design and characterization of knock-in lines at the *nkx6.1* locus.**
**(A)** The design of donor dsDNA template and gRNA sequence for the construction of knock-in lines at the 3′ end of the *nkx6.1* locus. The nucleotide sequence in blue indicates gRNA; whereas the nucleotide sequence in red indicates the PAM sequence. The bottom panel shows the sequences of LHA and RHA. **(B, C, D)** Summary statistics of *nkx6.1* knock-in efficiency, including the percentage of injected F0 with observable fluorescence labelling in the hindbrain and spinal cord (B), the percentage of adult F0 giving rise to germline transmission (out of those that bred) (C) and the percentage of F1 siblings inheriting each allele (D). **(E, F, G)** Representative confocal images of *TgKI(nkx6.1-p2A-mNeonGreen-t2A-iCre);Tg(ubi:CSHm)* at 1 (E), 2 (F), and 3 dpf (G). Cells expressing mNeonGreen indicate *nkx6.1⁺* cells; all progenies of *nkx6.1⁺* cells were labelled with H2BmCherry. The insets are magnified views showing the expression pattern of two fluorescent proteins. **(H)** Representative confocal image of lineage-tracing results in the principal islet of the pancreas in the *TgKI(nkx6.1-p2A-mNeonGreen-t2A-iCre);Tg(ubi:CSHm)* line at 6 dpf. Cells in white shown in H are β-cells with Insulin staining, whereas cells in green in H'' are α-cells with Glucagon staining. **(E, F, G, H)** Scale bars = 200 μm (E, F, G) or 20 μm (H).

injected embryos) *or nkx6.1-p2A-mNeonGreen* (in 4 out of 229 injected embryos) (Fig 2B). The percentages of founders among these mosaic F0 were between 75% and 100% (Fig 2C); 2.5–15.5% of the F1 siblings carried the knock-in cassettes (Fig 2D). For further comparison, we also injected *p2A-EGFP-t2A-CreERT2* dsDNA with short HAs and could identify 5 out of 155 (3.2%) injected embryos with detectable fluorescence. This indicated that short HAs are superior to (more than 10-fold) long HAs at this locus (Fig 2B) when comparing different donors that all carried the 5′ AmC6 modification. Sanger sequencing confirmed the correct in-frame integration at the junction between the endogenous gene and the 5′ end of the integrated sequences in all recovered *nkx6.1* lines (Fig S2 and data uploaded to a public repository: https://osf.io/tdkvh/).

The iCre and CreERT2 functions were characterized by the color switch in the offspring when crossed with *Tg(ubi:CSHm)*. We noticed that cells expressing *nkx6.1* (displayed by the green fluorescence) were located on the ventral side of the spinal cord; whereas H2BmCherry positive cells, which include all the progenies of *nkx6.1⁺* cells after the iCre recombination, resided in both the ventral and dorsal parts of spinal cord, suggesting a progenitor cell population of *nkx6.1⁺* cells in zebrafish spinal cord (Figs 2E–G and

S5A–C). In addition, a preceding immunostaining experiment using the *TgBAC(nkx6.1:EGFP)* reporter showed that *nkx6.1⁺* cells exist in both the dorsal and ventral buds of the pancreas at 17–48 hpf, indicating that they might be multipotent pancreatic progenitor cells (Binot et al, 2010; Ghaye et al, 2015); however, definitive evidence from lineage tracing experiments is still lacking. To further determine the *nkx6.1⁺* cell lineage in the pancreas, we firstly performed immunostaining for the fluorescent proteins in the knock-in lines and showed that *nkx6.1* specifically labelled the intrapancreatic duct in zebrafish larvae (Fig S5D–F). Secondly, using the *nkx6.1* knock-in iCre line, we could trace back all three major cell types in the pancreas (acinar, ductal, and endocrine cells) to *nkx6.1* lineage (Figs 2H–H''' and S5G–I), suggesting that all these different cell types were indeed derived from *nkx6.1⁺* cells.

## Short-term and long-term lineage tracing depicting the *nkx6.1* lineage

To further explore cell fate determination in the zebrafish pancreas, we did a lineage-tracing experiment using *TgKI(nkx6.1-p2A-EGFP-t2A-CreERT2);Tg(ubi:CSHm)* with 4-OHT treatments at multiple timepoints (Fig 3A). The immunostaining at 6 dpf showed that both

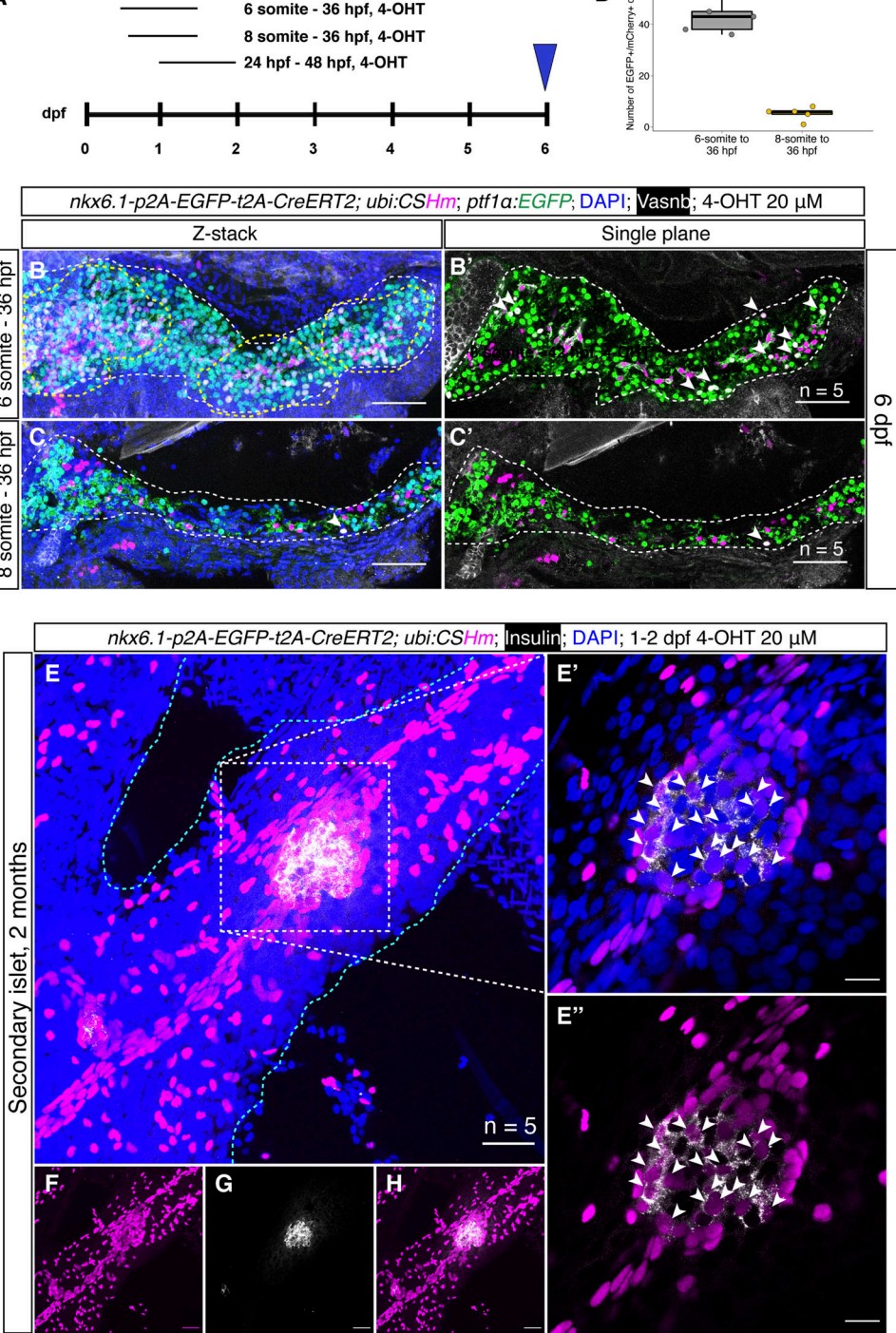

**Figure 3. nkx6.1⁺ cells were gradually restricted to the duct and gave rise to secondary islets in the zebrafish pancreas.**
**(A)** Experimental timeline for temporal labelling using short-term and long-term lineage tracing. 4-OHT (20 μM) was added at the 6 or 8 somite stage and the treatment continued until 36 hpf for short-term lineage tracing. **(B, C, E, F, G, H)** (B, B′, C, C′) 4-OHT (20 μM) was added from 1–2 dpf and confocal imaging was performed at 60 dpf for the long-term lineage tracing (E, F, G, H). Representative confocal images of the pancreas at 6 dpf in *TgKI(nkx6.1-p2A-EGFP-t2A-CreERT2; ptf1α:EGFP);Tg(ubi:CSHm)*. The progenies of *nkx6.1⁺* cells after 4-OHT treatment were H2BmCherry positive. The EGFP signals indicate acinar cells. Intrapancreatic ductal cells were demonstrated by membrane staining using the anti-Vasnb antibody (shown in white). The region within the yellow dashed line in B indicates H2BmCherry⁺/EGFP⁺ enrichment. Arrowheads in (B′, C, C′) point to H2BmCherry⁺/EGFP⁺/Vasnb⁻ cells, both indicating acinar cells from *nkx6.1⁺* cell origin. For each condition, we scanned five samples with 16–24 single-planes for larval pancreata and 18–30 single planes for juvenile pancreata. **(B, C)** The Z-stacked images were displayed (B, C) demonstrating a large number of mCherry⁺/EGFP⁺ cells with 4-OHT treatment starting from six-somite stage, whereas the number of double positive cells are decreased with statistical significance. **(D)** The quantification and statistical results of EGFP/mCherry double positive cells with 4-OHT treatment starting at 6 and 8 somite stages. Two-tailed *t* test was used for statistical analysis, with *P*-value < 0.05 considered as statistically significant. **(E, F, G, H)** Projection images of lineage-traced secondary islets in the *TgKI(nkx6.1-p2A-EGFP-t2A-CreERT2);Tg(ubi:CSHm)* zebrafish pancreas at 60 dpf. The progenies of *nkx6.1⁺* cells after the 4-OHT 1–2 dpf treatment were H2BmCherry positive. **(E)** The selected area in the white dashed square in (E) was magnified in a single plane (E′, E″). **(E)** The cyan dashed lines outline the pancreas (E). **(E, F, G, H)** Split channels of (E) are displayed (F, G, H). Arrowheads point to lineage-traced β-cells in the secondary islet co-stained with an anti-Insulin antibody. **(B, C, E, F, G, H)** Scale bars = 80 μm (B, B′, C, C′), 20 μm (E, F, G, H), or 10 μm (E′, E″), respectively.

intrapancreatic ductal cells and a portion of acinar cells can be lineage traced when the 4-OHT treatment started at the 6-somite stage (Fig 3B and B′). In contrast, *nkx6.1* lineage-traced cells were mostly restricted to the intrapancreatic duct when the 4-OHT treatment started at the eight-somite stage (Fig 3C, C′, and D). These results pinpoint the exact timing of the early cell fate divergence between acinar cells and ductal/endocrine lineages.

Previous studies using transgenic lines based on *tp1* promoter, which is a Notch-responsive element from the Epstein–Barr virus mediating expression in the intrapancreatic duct (including *Tg(tp1: H2Bmcherry)*, *Tg(tp1:venusPEST)*, and *Tg(tp1:CreERT2)*), suggested that Notch-responsive intrapancreatic ductal cells can give rise to endocrine cells. These endocrine cells are mainly located in secondary islets and appear during growth or upon Notch inhibition (Parsons et al, 2009; Ninov

et al, 2013; Delaspre et al, 2015). In addition, previous immunostaining results in *TgBAC(nkx6.1:EGFP)* showed insulin/EGFP colocalization in adult zebrafish pancreatic tail region after *β*-cell ablation, suggesting latent duct-to-*β*-cell neogenesis is maintained in adulthood (Ghaye et al, 2015; Carril Pardo et al, 2022). However, lineage tracing using *tp1:CreERT2* can only label at maximum 75% of the Notch-responsive ductal cells and traced a very limited amount of endocrine cells, indicating that *tp1* was not an efficient tracer for ductal neogenesis (Delaspre et al, 2015; Singh et al, 2017). To have a better understanding of duct-to-*β*-cell neogenesis, we performed a 2-month long-term lineage-tracing experiment using *TgKI(nkx6.1-p2A-EGFP-t2A-CreERT2);Tg(ubi:CSHm)* with 4-OHT treatment at 1–2 dpf. We observed that nearly 65% of *ins*[+] cells in the secondary islets (residing along the large ducts) can be lineage traced (Figs 3E–H and S7D). Furthermore, we performed short-term lineage-tracing experiments in the presence of a Notch inhibitor (LY411575) or a REST inhibitor (X5050), as previous studies described (Ninov et al, 2012; Rovira et al, 2021), to demonstrate the neogenic potential of *nkx6.1*[+] intrapancreatic ductal cells in zebrafish larvae (Fig S6). We found that more than 80% of *ins*[+] cells and around 75% of glucagon positive (*gcg*[+]) cells in the pancreatic tail can be lineage traced after 3-d treatment (3–6 dpf) with LY411575 and X5050, respectively. These results, all together, suggested a latent progenitor

property of the *nkx6.1*[+] duct, one that can serve as the origin for endocrine cells.

Lastly, regarding the CNS, the temporal-controlled experiments showed a similar labelling pattern to the noninducible Cre *nkx6.1* knock-in line, confirming that the *nkx6.1*[+] cells in the spinal cord are also progenitors (Fig S7A–C). Control experiments without 4-OHT treatment indicated no leakage problem with the *nkx6.1* knock-in CreERT2 line (Fig S8).

### Generation of *krt4* knock-in lines using 5′ modified dsDNA with short HAs

Similar to *nkx6.1*, we identified one gRNA target spanning over the stop codon in *krt4* (Fig 4A). To assess the knock-in efficiency of a dsDNA donor using short HAs with 5′ modifications, we amplified three different insertion sequences (*p2A-mNeonGreen*, *p2A-mNeonGreen-t2A-iCre*, and *p2A-EGFP-t2A-CreERT2*) with pairs of primers containing 32 and 33 base pairs homologous overhang at the 5′ ends by PCR (Fig 4A). Around 2.0–7.4% of injected F0 displayed green fluorescence labelling in at least 30% of the skin at 1 dpf (Fig 4B). In addition, more than 75% of these F0 were characterized as founders (Fig 4C). The proportion of the F1 generation that carried the knock-in cassettes ranged from 1.2–22.0% (Fig 4D). The iCre

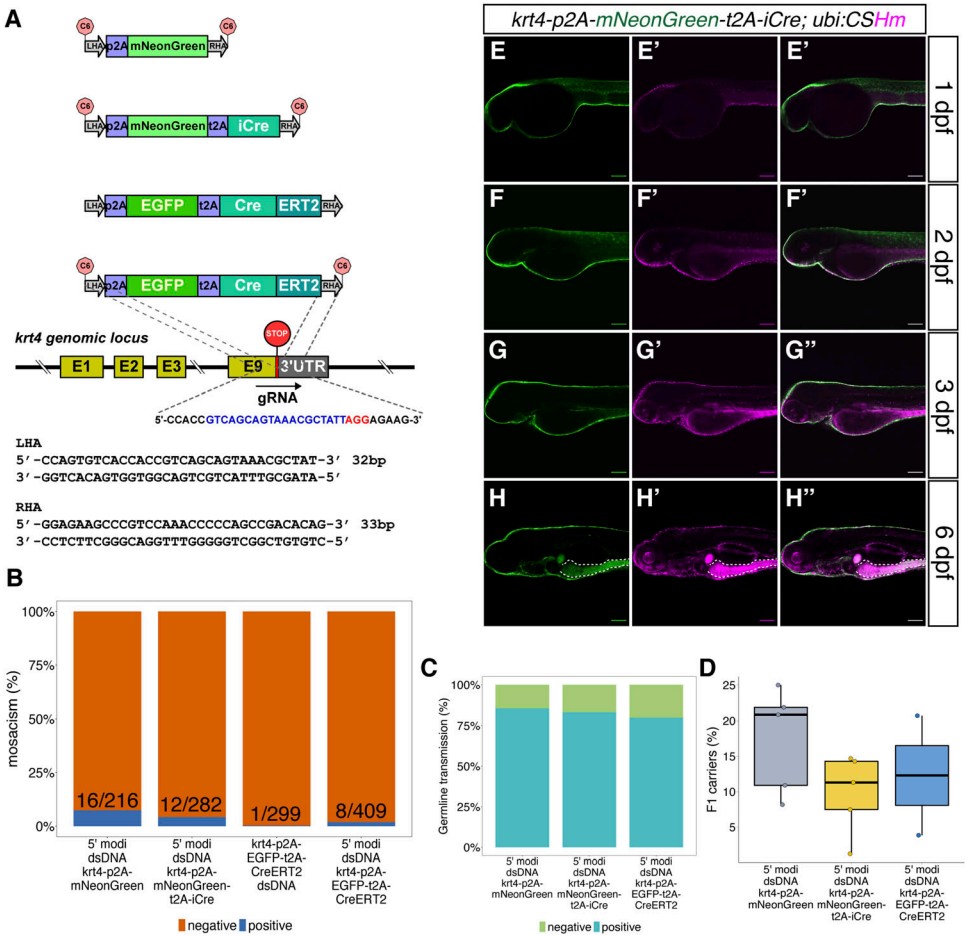

**Figure 4. The design and characterization of knock-in lines at the *krt4* locus.**
**(A)** The design of donor dsDNA templates and gRNA sequence for the construction of knock-in lines at the 3′ end of the *krt4* locus. The nucleotide sequence in blue indicates gRNA, whereas the nucleotide sequence in red indicates the PAM sequence. The bottom panel shows the sequences of LHA and RHA. **(B, C, D)** Summary statistics of knock-in efficiency at the *krt4* locus, including the percentage of injected F0 with fluorescence labelling on approximately one-third of their skin. **(B, C, D)** Knock-in mosaicism (B), the percentage of adult F0 giving rise to germline transmission (C), and the percentage of F1 siblings carrying the knock-in cassettes (D). **(E, F, G, H)** Representative confocal images of *TgKI(krt4-p2A-mNeonGreen-t2A-iCre); Tg(ubiCSHm)* at 1 dpf (E, E′, E″), 2 dpf (F, F′, F″), 3 dpf (G, G′, G″), and 6 dpf (H, H′, H″). Skin and intestinal epithelial cells (at 3 and 6 dpf) were broadly recombined and labelled with H2BmCherry. **(H)** The white dashed lines outline the intestinal bulb (H, H′, H″). Scale bars = 80 *μ*m.

function was confirmed by crossing the *TgKI(krt4-p2A-mNeon-Green-t2A-iCre)* with *Tg(ubi:CSHm)*. We observed H2BmCherry labelled cells in the skin and intestine (Fig 4E–H). However, *TgKI(krt4-p2A-EGFP-t2A-CreERT2);Tg(ubi:CSHm)* embryos showed mosaic leakage in the skin and intestine without 4-OHT treatment (Fig S9A–C). For systematic comparison, we also injected a *p2A-EGFP-t2A-CreERT2* PCR-amplified donor without end protection and observed that only 0.3% (1 out of 299) injected F0 achieved early integration based on the above criteria (Fig 4B), indicating that the 5′ modification can achieve around fivefold more efficient integration of dsDNA donors when using short HAs at this locus. Lastly, to compare the knock-in efficiency of different modifications, that is, AmC6 versus biotin, which has previous been used as dsDNA end protection for making knock-in lines in medaka, we injected the same batch of embryos with each 5′ modified donors. In the F0, we observed 35 out of 428 embryos displayed >30% skin area with green fluorescence using the AmC6 modification, whereas only 3 out of 526 embryos displayed visible green using the biotin modification, indicating that at least in certain genomic locus, the AmC6 modification results in better integration efficiency than biotin. We also confirmed the correct in-frame integration at the 5′ end of the integrated sequences in all the recovered *krt4* lines using Sanger sequencing (Fig S2 and data uploaded to a public repository: https://osf.io/tdkvh/).

As the *krt4* transgenics have been widely used for labelling skin epithelial cells (Lam et al, 2013; Fischer et al, 2014) (Fig S9D), we performed a further characterization of *krt4* expression in the gut using HCR3.0 in situ hybridization and EGFP immunofluorescence in both *Tg(krt4-p2a-EGFP-t2a-CreERT2)* and *Tg(krt4:EGFP-Mmu.Rpl10a)* zebrafish larvae (Fig S10). Notably, the *krt4* in situ result showed very strong signals in both the intestinal bulb and the hindgut. The green fluorescence in the *krt4* knock-in line fully recapitulates endogenous *krt4* expression (Fig S10C and D), whereas the *krt4* transgenics displayed no signals in the gut (Fig S10A and B), indicating that the *cis*-regulatory element cloned in the *krt4* transgenic line is unable to mimic endogenous *krt4* expression and insufficient to drive EGFP expression in the gut.

## Generation of *id2a* knock-in lines using short HAs

Next, we used similar strategies to knock-in *p2A-mNeonGreen*, *p2A-mNeonGreen-t2A-iCre*, and *p2A-EGFP-t2A-CreERT2* fragments into the 3′ end of the *id2a* locus using short HAs (Fig 5A). We found that 1.5% (2 out of 137), 0.8% (2 out of 252), and 3.0% (7 out of 240) injected embryos displayed strong fluorescence in the hindbrain, spinal cord, and olfactory organs and were raised up to adulthood (Fig 5B). We observed high percentage of mosaic F0 with germline transmission (100%, 100%, and 71.4%, respectively) (Fig 5C) and the percentage of the F1 generation carrying the cassettes ranged from 3.2–29.0% (Fig 5D). We crossed the iCre and CreERT2 knock-in lines with *Tg(ubi:CSHm)* and observed a similar pattern of mCherry signal as mNeonGreen in various tissues (hindbrain, dorsal side of spinal cord, pronephros, olfactory organ, and muscles), verifying the functionality of the Cre recombinases (Figs 5E–G and S11A–C). Control experiments suggested that there is no leakage problem with the *id2a* knock-in CreERT2 line (Fig S12). Sanger sequencing confirmed the correct in-frame integration at the junction between

the endogenous gene and the 5′ end of the integrated sequences in all recovered *id2a* lines (Fig S2 and data uploaded to a public repository: https://osf.io/tdkvh/).

Previous studies have shown that *id2a* is important in the development of endodermal organs (e.g., liver and pancreas) (Khaliq et al, 2015; Choi et al, 2017; Tarifeno-Saldivia et al, 2017) and retina (Uribe & Gross, 2010). Therefore, we performed immunostaining of the knock-in *EGFP* line to visualize *id2a* expression in these tissues. In the zebrafish gut (intestinal bulb and hindgut), we observed a large number of *id2a*[+] cells showing overlapping fluorescence signals with *tp1:H2BmCherry*, suggesting that these cells are *id2a*[+] and have active Notch signaling (Fig S11D and E). These double positive cells had apical membranes oriented towards the gut lumen, indicating that they were either sensory cells or secretory cells (Fig S11F). This result was in line with two recent single-cell RNA-seq studies of the larval gut, proposing the subpopulation of Notch-responsive cells were *best4/otop2*[+] ionocytes (Fig S13E–H) (Wen et al, 2021), which regulate ion concentrations.

In the zebrafish pancreas and liver, the *id2a* knock-in reporter showed fluorescence in the intrapancreatic duct and a subset of extrahepatic and intrahepatic ductal cells, and in what we term the intermediate duct that branches out in between the extrahepatic and intrahepatic ducts (Figs S11G–J and S13A–D). In the retina, the *id2a* reporter labelled a substantial number of retinal epithelial cells (Fig S11K and L). Altogether, the *id2a* knock-in reporter closely recapitulated known expression domains of *id2a*.

## id2a tracing delineates developmental paths in the liver and pancreas

Previous fate mapping studies suggested that the pancreas and liver originate from common progenitors in the endodermal sheet, in which cells close to the midline are prone to differentiate into pancreatic endocrine cells and intestine; whereas cells located two cells away from the midline are inclined to develop into liver and exocrine pancreas (Chung et al, 2008; Yang et al, 2021). The mediolateral patterning of cell fate decision relies on mesoderm-derived Bmp2b (Chung et al, 2008). Higher levels of Bmp2b promote liver versus pancreas development, whereas notochord-derived Nog2, a Bmp antagonist, increases the number of pancreatic progenitors (Amorim et al, 2020). Given that *id2a* belongs to the inhibitor of a differentiation protein family and is a downstream gene of Bmp signaling, we use both *id2a* iCre and CreERT2 knock-in lines for lineage tracing to have a better understanding of the hepatic–pancreatic development. The results from *TgKI(id2a-p2A-mNeonGreen-t2A-iCre);Tg(ubi:CSHm)* demonstrated a universal labelling in the liver and pancreas, suggesting that *id2a*[+] cells are common hepatic-pancreatic multipotent progenitors (Fig 5H and I). Furthermore, we did temporal labelling by treating with 4-OHT at several timepoints (20, 24, 32, and 48 hpf) for 24 h (Fig 6A–D). Both hepatocytes and hepatic ducts were H2BmCherry positive when labelled at 20 hpf (Fig 6A). However, the H2BmCherry positive cells in the liver were gradually restricted to the hepatic duct at 24 hpf (Fig 6B) and demonstrated specific hepatic duct labelling when the treatment started at 32 or 48 hpf (Figs 6C and D and S14). In the zebrafish pancreas, similar to *nkx6.1*, temporal labelling of *id2a*[+] cells at 1 dpf marked mainly ductal cells (intra- and extra-

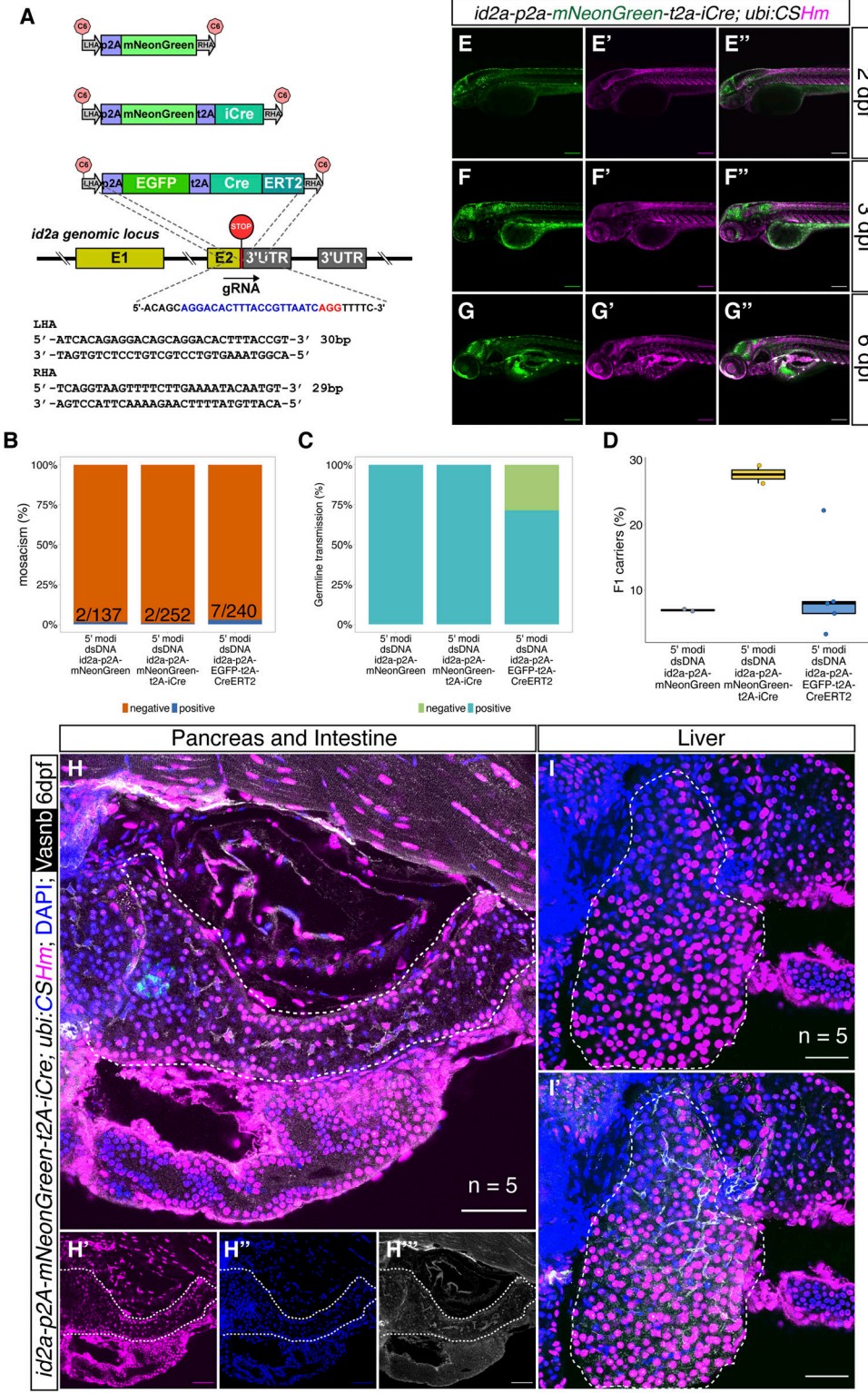

**Figure 5. The design and characterization of knock-in lines at the *id2a* locus.**
**(A)** The design of the donor dsDNA template and gRNA sequence for the construction of knock-in lines at the 3′ end of the *id2a* locus. The nucleotide sequence in blue indicates gRNA, whereas the nucleotide sequence in red indicates the PAM sequence. The bottom panel shows the sequences of LHA and RHA. **(B, C, D)** Summary statistics of *id2a* knock-in efficiency, including the percentage of injected F0 with observable fluorescence labelling in the hindbrain, spinal cord, and olfactory organs (B), the percentage of adult F0 giving rise to germline transmission (C), and the percentage of F1 siblings carrying the knock-in cassettes (D). **(E, F, G)** Representative confocal images of *TgKI(id2a-p2A-mNeonGreen-t2A-iCre); Tg(ubi:CSHm)* at 2 dpf (E, E′, E″), 3 dpf (F, F′, F″), and 6 dpf (G, G′, G″). Cells that are mNeonGreen positive indicate *id2a*-expressing cells, whereas the progenies of the *id2a* lineage were mCherry labelled. **(H, I)** Representative confocal images of lineage-tracing experiments in the zebrafish larval pancreata (H, H′, H″, H‴), intestine (H, H′, H″, H‴), and liver (I, I′) in the *TgKI(id2a-p2A-mNeonGreen-t2A-iCre);Tg(ubi:CSHm)* line. We scanned five pancreata and livers, with 26–34 single planes imaged. Cells in cyan are α-cells based on anti-Glucagon antibody staining. **(E, F, G, H, I)** Scale bars = 200 μm (E, F, G) or 80 μm (H, I).

pancreatic ductal cells) and a subpopulation of endocrine cells (Fig 6E and F). These results suggest that Bmp signaling is active in the liver, dorsal bud-derived pancreas, and ventral bud-derived pancreas specification at early developmental stages, whereas its activity is preferentially maintained in hepatic and pancreatic duct.

Lastly, we examined whether there is ductal-to-hepatocyte conversion in two liver injury models, as previous studies showed

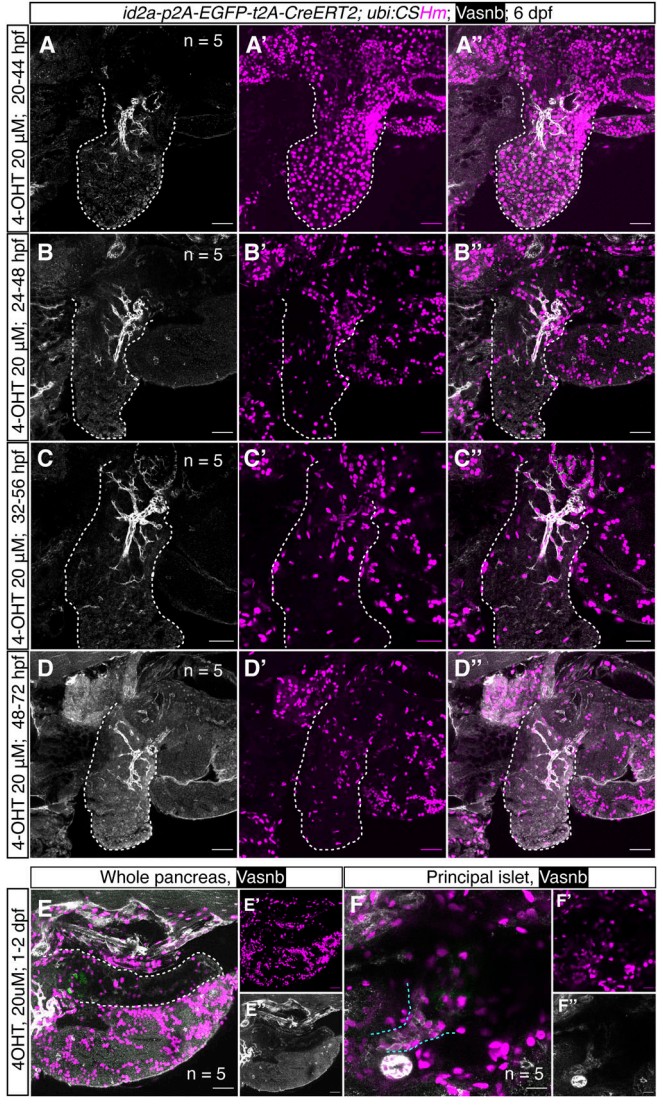

**Figure 6. Temporally controlled _id2a_ lineage-tracing experiments in the zebrafish liver and pancreas.**
**(A, B, C, D)** Representative confocal images of the liver in _TgKI(id2a-p2A-EGFP-t2A-CreERT2)_ treated with 20 μM 4-OHT at 20 hpf (A, A', A''), 24 hpf (B, B', B''), 32 hpf (C, C', C''), and 38 hpf (D, D', D'') for 24 h. For each condition, we scanned five embryos; for each embryo, 20–32 single planes were imaged. The progenies of _id2a_⁺ cells were labelled with H2BmCherry. The white dashed lines indicate the liver.
**(E, F)** Representative confocal images of zebrafish pancreas and principal islet at 6 dpf treated with 4-OHT 20 μM at 1–2 dpf. (F', F'') Magnified confocal images of (F) showing zebrafish principal islets. Extrapancreatic ductal cells and the principal islet are defined by anti-Vasnb antibody (in white) and anti-Glucagon antibody (in green), respectively. The white dashed lines indicate the pancreas; the cyan dashed lines indicate the extrapancreatic duct. **(A, B, C, D, E, F)** Scale bars = 35 μm (A, B, C, D), 40 μm (E) or 20 μm (F).

that _tp1_⁺ intrahepatic ductal cells can convert to hepatocytes in an extreme hepatocyte ablation condition (He et al, 2014). Here, we changed the responder line to _Tg(−3.5ubb:loxP-EGFP-loxP-mCherry)_ (abbreviated as _Tg(ubi:Switch)_) for the lineage-tracing experiment, as the cytoplasmic mCherry can allow us to distinguish the cell type based on morphology (Choi et al, 2014). The _id2a_⁺ hepatic ductal cells were first temporally labelled from 2–3 dpf,

with subsequent metronidazole (MTZ) treatment using larvae carrying the _fabp10:CFP-NTR_ transgene or with chemical-induced severe liver injury (acetaminophen 10 mM + 0.5% ethanol for 48 h, resulting in near 90% hepatocyte ablation) from 3–5 dpf followed by 2 d of regeneration (North et al, 2010). Based on the morphology of the mCherry positive cells and the Vasnb co-staining, we confirmed that a large number of hepatocytes were derived from an _id2a_⁺ duct origin after the MTZ/NTR-induced liver injury (Fig 7). All lineage-traced cells, however, retained their ductal identity after the chemical-induced injury (Fig S15), indicating that _id2a_⁺ duct-to-hepatocyte conversion mainly occurs after extreme hepatocyte loss, whereas this phenomenon is very sparse in the severe liver injury model. Thus, the Notch- and BMP-responsive hepatic ductal cells maintain progenitor potential in zebrafish.

## Discussion

Here, we introduce a straightforward 3′ knock-in pipeline to generate zebrafish lines for both cellular labelling and lineage tracing. This method combines a one-step PCR amplification for 5′ modified dsDNA donor with short or long HAs and coinjection with Cas9 protein/gRNA RNPs. Notably, we observed high F0 mosaic integration (often half of the injected embryos displayed some fluorescence/integration, but it varied depending on the target) and a very high germline transmission rate of those with >30% mosaicism, indicating that this method can achieve early genetic integration. By systematic comparisons with different donors, we proposed that 5′ modified dsDNAs with short HAs had the best performance when gRNAs spanning over stop codons are available. Lastly, we managed to knock in large DNA fragments with multiple cassettes linked by 2A peptides that generated zebrafish lines useful for multiple applications. In all, this straightforward and highly efficient knock-in pipeline is versatile and amenable to a wide range of users, allowing researchers to carry out knock-in projects in a scalable fashion.

The CRISPR-Cas9 mediated knock-in method for experimental animals was first introduced in mouse models and, afterwards, optimizations have been developed in various organisms. However, each optimization approach needs to be rigorously tested in each individual model because of their huge differences in early embryonic development. The NHEJ-guided 5′ knock-in upstream of ATG, NHEJ-guided intron-based knock-in, exon-based knock-in (such as the Geneweld toolbox), and 3′ knock-in with circular plasmids have been reported to generate several reporter lines and floxed lines (Auer et al, 2014). In addition, several recent studies used dsDNA as the direct donor. The Tild-CRISPR method, which is based on in vitro vector linearization by PCR amplification or enzyme cutting, can dramatically increase the knock-in efficiency in mice (Yao et al, 2018b). In zebrafish, however, there are conflicting results: one former study showed that such in vitro linearization was inefficient to mediate HDR compared with circular plasmids, whereas another study reported that coinjection of synthetic gRNA and linear dsDNA together showed promising results (Hoshijima et al, 2016; DiNapoli et al, 2020). In addition, PCR amplicons containing fluorescent tagging with several hundred base pairs

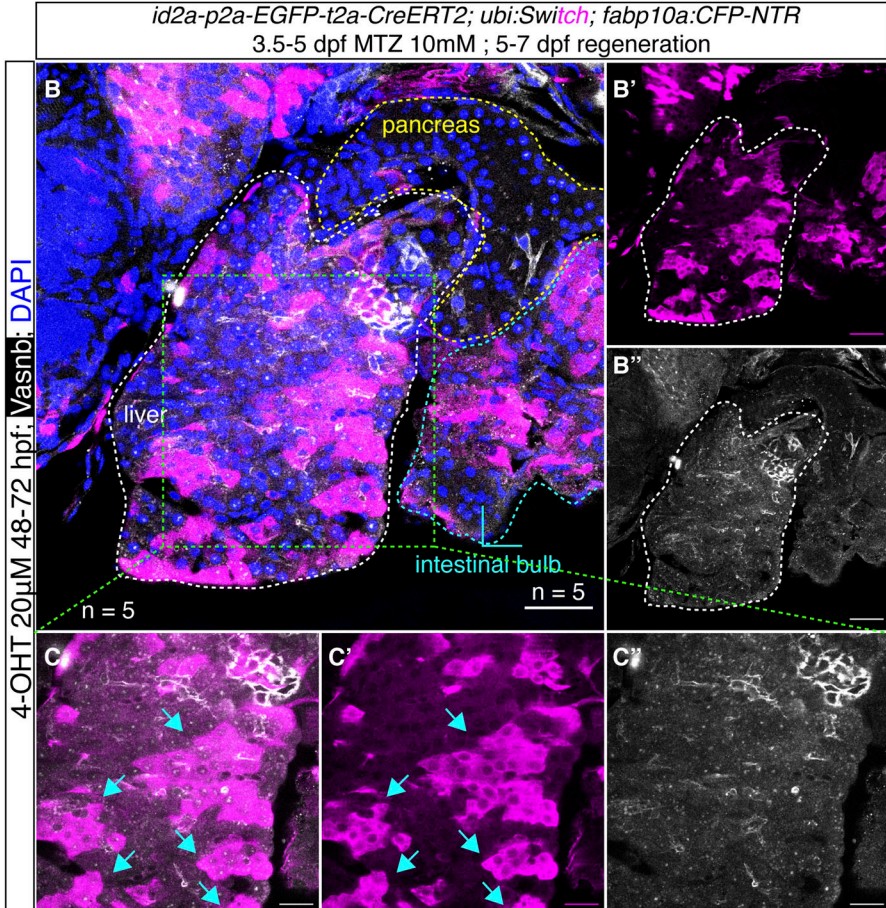

**Figure 7. *id2a* lineage-traced cells in an extreme liver injury model.**
**(A)** Experimental timeline of *id2a* lineage tracing in an MTZ/NTR-induced extreme liver injury model. **(B)** Representative confocal images of *id2a* lineage-traced cells in the extreme liver injury model. In total, we scanned five samples with 21–35 single planes in each zebrafish larvae. The white dashed lines indicate the liver, the yellow dashed lines indicate the pancreas, and the cyan dashed lines indicate the intestinal bulb. **(C)** Magnified image of the liver showing large numbers of regenerated hepatocytes lineage-traced back to an *id2a*⁺ cellular origin. The arrows point to clusters of regenerated hepatocytes. **(B, C)** Scale bars = 40 *μm* (B) and 20 *μm* (C).

flanking sequences can be successfully knocked in to noncoding regions at N- or C-terminal regions, indicating that PCR amplicons can be useful donors in facilitating zebrafish HDR-based knock-in (Levic et al, 2021). Interestingly, one study systematically compared 13 modifications on dsDNA donors with short HAs in generating knocked-in human cell lines and showed several modifications (especially 5′ C6-PEG10, amine group with a C6 linker [5′ AmC6] or C12 linker [5′ AmC12]) can exceptionally enhance the 3′ knock-in rate (with up to 500%) (Yu et al, 2020). Although the underlying mechanism is not fully understood, it is proposed that the 5′ end protection can prevent the NHEJ event and donor multimerization (Gutierrez-Triana et al, 2018).

Here, we focused on 3′ knock-in because this method, theoretically, can keep the endogenous gene functional and intact. We observed that short HAs were sufficient when knocking in long DNA fragments, that is, when there is a good gRNA spanning over the stop codon region. Furthermore, our data targeting the *krt92* and *krt4* loci suggesting that AmC6-end protection greatly improves the efficiency of HDR integration in zebrafish are consistent with the idea that end protection of dsDNA limits exonuclease degradation and concatenation. Given that the dsDNAs with short HAs are smaller in size (compared with circular plasmid or dsDNA with long HAs), we assume that they can more efficiently translocate into the nucleus. Altogether, the 5′ end protection and short HAs may lead to a high concentration of dsDNA in the local integration region for the promotion of HDR. In *krt4* locus, we also observed that the AmC6-end protection showed much higher efficiency than biotin-end protection. Although it is hard to conclude that the AmC6 modification is generally better than biotin in zebrafish, we showed that different modifications can impact the integration efficiency and converting to alternative modifications could be a simple way to improve knock-in efficiency in certain genetic loci rather than switching to a different gRNA. However, we believe it is difficult to make comparisons regarding efficiency between various knock-in methods because different loci were chosen for testing (Irion et al, 2014; Hoshijima et al, 2016; DiNapoli et al, 2020; Almeida et al, 2021).

The main advantages of our method become apparent in several aspects. First, we are targeting the 3′ end without the disruption of the endogenous gene products. In addition, the 3′ in-frame integration did not show any effects on the mRNA expression of the endogenous genes (Fig S16). This is of great importance in developmental and regenerative studies as in certain cases, the loss of one gene allele (i.e., in transcription factors) can generate detectable phenotypes (Delous et al, 2012). Second, we found that dsDNAs with 5′ modifications were efficient donors in zebrafish. Moreover, the short HAs were at least as equally good as long HAs in the locus we tested, although that could depend on the intrinsic features of these particular long HAs (e.g., structure, repeats, etc.). However, the main point is that short HAs are sufficient and the preparation of donors can be dramatically simplified by one-step PCR using primers harboring 29–41 base pairs overhangs, which is in line with previous work (Luo et al, 2018). Third, the design of 2A peptides linking different functional cassettes enables us to generate knock-in lines for multiple uses. Fourth, it is much easier to identify the mosaic founders for knock-in of nonfluorescent protein (e.g., Cre) when combined with a fluorescent protein as an in-frame marker (which also indicates correct integration). Fifth, although previous studies supported the use of Cas9 mRNA in medaka (Gutierrez-Triana et al, 2018; Seleit et al, 2021), here, we instead recommend injecting Cas9 protein in zebrafish embryos to enable early integration. Medaka has a much slower cell cycle during early development (16-cell stage at 3 hpf) (Iwamatsu, 2004), whereas zebrafish embryos display very rapid cell division (over 1,000 cells at 3 hpf) (Kimmel et al, 1995). The generation of dsDNA breaks need to be concordant with a sufficient amount of donor templates ready for HDR, otherwise, cells are inclined to use error-prone NHEJ, which would introduce indels. As Cas9 protein can be transported into the nucleus rapidly, it can cleave genomic DNA soon after injection, which is particularly important for obtaining high germline transmission in zebrafish. Given that dsDNAs are smaller in size and can diffuse faster than whole plasmids, the cleaved genomic DNA has a higher chance to be precisely repaired by MMEJ rather than NHEJ. Sixth, we chose the strongest green fluorescence monomer, mNeonGreen, as an alternative to EGFP for cellular labelling. Strong fluorophores are helpful for the identification of mosaic embryos in case the targeted gene has a low expression. Moreover, the mNeonGreen is not recognized by the anti-GFP antibody, allowing the users to combine it with transgenic lines expressing GFP or its derivatives.

We shall also note that the use of short HAs is highly dependent on whether there is a good gRNA spanning over the stop codon region. If there is an absence of such gRNA, long HAs might be a better choice as mutations need to be introduced in the left (for gRNA sites upstream of the stop codon) or right (for gRNA sites downstream of the stop codon) HAs, such that the gRNA does not also cleave the donor dsDNA. However, the use of short HAs provides an easy and efficient way to knock in a specific donor into several candidate genes in parallel, that is, without needing to perform complex molecular cloning steps.

We also employed our knock-in lines to investigate cell fate determination in the liver and pancreas. We, for the first time, used a lineage-tracing method to make a definitive conclusion that the *nkx6.1*⁺ intrapancreatic ductal cells can serve as progenitor cells that can differentiate into endocrine cells in secondary islets. Also, by combining the lineage-tracing results from noninducible and inducible Cre lines, we depicted a temporal cell differentiation map for the specification between the acinar fate and ductal/endocrine fate in the pancreas, and between hepatocyte fate and hepatic ductal fate in the liver. Lastly, we consolidated and extended the previous findings indicating that Notch- and BMP-responsive liver ductal cells are progenitors and able to convert to hepatocytes only upon extreme liver injury (Choi et al, 2014; He et al, 2014). Further experiments using lineage tracing, single-cell RNA sequencing, and tissue-specific gain-and-loss of function are warranted to investigate detailed cellular and molecular mechanisms in the development and regeneration of these tissues.

Through systematic comparisons of the pattern of *krt4* gene expression in both widely used transgenics and our knock-in lines, we found that our knock-ins can fully recapitulate the endogenous gene expression, whereas the *krt4 cis*-regulatory element cloned for skin cell labelling is incapable of driving the reporter gene expression in the intestine (intestinal bulb and hindgut). This is of particular importance for lineage-related research as most previous cell lineage discoveries in zebrafish are based upon cloning promoters for transgenics, which may fail to label certain cell types, display ectopic expression or exhibit leakage issues. This might be because of the fact that different tissues/cell types tend to use different *cis*-regulatory elements with different chromatin structures, and that the enhancer–promoter loops might vary greatly in different cell types (Heinz et al, 2015). It is difficult to precisely predict the exact region of the regulatory sequences sufficient to activate the gene expression in each cell type without a comprehensive profiling of the epigenetic landscape in a single-cell resolution. Therefore, we believe that our method, together with other zebrafish knock-in Cre/CreERT2 methods, might divert the standard toward knock-in-based genetic fate mapping for both confirming old and making new discoveries.

We should note that our knock-in strategy is a good alternative to the current zebrafish genome editing toolbox and can complement other methodologies depending on the purpose of the research. Our method is useful for generating knock-in tracers to delineate natural occurring events without disruption of the endogenous gene product. However, other knock-in strategies, especially the generation of knock-in/knock-out lines can greatly help to trace cells in loss-of-function conditions. We also suggest users to target genes with strong and widespread expression patterns with our method as genes with low or very restricted expression patterns might not be visible even if you achieve integration at the single-cell stage. Further studies combining our method with methods involving a sorting marker (with fluorescence in the eye or heart) could be a promising strategy to target such genes (Wierson et al, 2020; Tan & Winkler, 2022).

In summary, we described a novel 3′ knock-in pipeline for the construction of zebrafish lines with multiple cassettes. This method is easy to implement as it only includes a one-step PCR reaction and coinjection with Cas9/gRNA RNPs. The application of dsDNA with short HAs allows us to knock-in specific donors in multiple genes in a scalable fashion. This method is highly efficient as it can achieve a desirable percentage of germline transmission from mosaic F0, which is helpful for small-to-medium sized zebrafish

labs with limited space to screen for founders. The design using 2A peptides for linkage makes it possible to knock-in multiple cassettes at the same genetic locus, further expanding the utility of the knock-in lines. Therefore, we anticipate that this efficient and straightforward knock-in method will be of widespread use in the zebrafish field.

# Materials and Methods

### Zebrafish lines used in the study

Males and females ranging from 3 mo to 2 yr were used for breeding. Zebrafish larvae were incubated in 28.5°C until 7 dpf. The following published transgenic zebrafish (*Danio rerio*) lines were used in this study: *Tg(ptf1a:GFP)*[jh1] (Godinho et al, 2005), *Tg(Tp1bglob: H2BmCherry)*[S939] (Ninov et al, 2012) abbreviated *Tg(Tp1:H2BmCherry)*, *Tg(−3.5ubb:loxP-EGFP-loxP-mCherry)*[cz1701] (Mosimann et al, 2011) abbreviated as *Tg(ubi:Switch)*, *Tg(UBB:loxP-CFP-STOP-Terminator-loxP-hmgb1-mCherry)*[jh63] (Zhang et al, 2017) abbreviated as *Tg(ubi:CSHm)*, *Tg(ela3l:H2BGFP)* (Schmitner et al, 2017), *Tg(krt4:EGFP-Mmu.Rpl10a)* (Lam et al, 2013), and *Tg(fabp10a:CFP-NTR)*[S931] (Choi et al, 2014).

The following lines were newly generated by the CRISPR-Cas9 3′ knock-in strategy: *TgKI(krt92-p2A-EGFP-t2A-CreERT2)*[KI126], *TgKI(krt4-p2A-mNeonGreen)*[KI127], *TgKI(krt4-p2A-mNeonGreen-t2A-iCre)*[KI128], *TgKI(krt4-p2A-EGFP-t2A-CreERT2)*[KI129], *TgKI(nkx6.1-p2A-mNeonGreen)*[KI130], *TgKI(nkx6.1-p2A-mNeonGreen-t2A-iCre)*[KI131], *TgKI(nkx6.1-p2A-EGFP-t2A-CreERT2)*[KI132], *TgKI(id2a-p2A-mNeonGreen)*[KI133], *TgKI(id2a-p2A-mNeonGreen-t2A-iCre)*[KI134], and *TgKI(id2a-p2A-EGFP-t2A-CreERT2)*[KI135].

Adult fish were maintained on a 14:10 light/dark cycle at 28°C. All studies involving zebrafish were performed in accordance with local guidelines and regulations and were approved by the ethical committee (called Stockholms djurförsöksetiska nämnd).

### The vector design for 3′ knock-in

The vector templates were manually designed for *krt92* and *nkx6.1* 3′ end loci. The vectors include a left long HA of 900 base pairs genomic sequence upstream of the stop codon of the endogenous gene product followed by a GSG linker (glycine–serine–glycine, GGAAGCGGA), p2A sequence (GCTACTAACTTCAGCCTGCTGAAG-CAGGCTGGAGACGTGGAGGAGAACCCTGGACCT), zebrafish codon optimized EGFP or mNeonGreen (without stop codons), GSG linker, t2A sequence (GAGGGCAGAGGCAGTCTGCTGACATGCGGTGATGTGGAAGA-GAATCCCGGCCCT), zebrafish codon optimized iCre or CreERT2 (with stop codons), and 950 base pairs right long HA downstream of endogenous stop codon flanked by in vivo linearization sites (GAGCTCGGTACCCGGGGATC[AGG] on the left; ATCCTCTAGAGTCGACCTGC [AGG] on the right).

### PCR amplification and gel purification

The 5′ modified primers were ordered with AmC6 5′ modification from Integrated DNA Technologies. The primer powders were diluted with distilled water into 100 mM as stock solution. The 50 µl PCR mixture includes:

Forward primer: 2.5 µl
Reverse primer: 2.5 µl
Template plasmid: 1 µl
Distilled water: 19 µl
Q5 Hot Star High-Fidelity (Hifi) 2× Master Mix: 25 µl
We use the following PCR cycle setting:
Pre-denaturing: 98°C, 30 s
Denaturing: 98°C, 10 s
Annealing: 58–60°C, 20 s
Extension: 72°C, 90 s
Final extension: 72°C, 2 min, and hold at 4°C

Next, we ran the PCR products in 1% agarose gel with 100 V for 45–60 min. The corresponding bands were cut out and purified using the wizard SV gel and PCR clean-up system (A9282; Promega). The concentrations of purified PCR products were measured by NanoDrop (2000c) and then diluted with distilled water to 70–100 ng/µl. The purified PCR products were stored at –20°C before injection.

### The selection of gRNA

We used the CHOPCHOP web-based tool (http://chopchop.cbu.uib.no/) and set the reference genome as "danRer11/GRCz11." We selected "CRISPR/Cas9" and the "knock-in" module after determining the targeted gene. This tool would scan through the gene exon regions and rank the gRNA based on efficiency score, self-complementarity, and the number of mismatches. Apart from the in silico prediction, we also manually examined the 3′ end in the Ensembl database (https://www.ensembl.org/Danio_rerio/Info/Index) to avoid polymorphisms in the targeting site. The following gRNAs were used in this study (with PAM sequence in the brackets):

*krt92* (on reverse strand): 5′-AACCTCGCTCGAGATTGGG(AGG)-3′
*krt4* (on forward strand): 5′-GTCAGCAGTAAACGCTATT(AGG)-3′
*nkx6.1* (on forward strand): 5′-AGAGCTCGTAAAAAGGAAAC(GGG)-3′
*id2a* (on forward strand): 5′-AGGACACTTTACCGTTAATC(AGG)-3′

For the control experiment using plasmids with linearization sites, we used the following gRNAs:

Left: 5′-GAGCTCGGTACCCGGGGATC(AGG)-3′
Right: 5′-ATCCTCTAGAGTCGACCTGC(AGG)-3′.

### In vitro preassembly of gRNA, Cas9 protein, and donor dsDNA

The chemically synthesized Alt-R-modified crRNA, tracrRNA, Hifi Cas9 protein, and nuclease-free duplex buffer were ordered from Integrated DNA Technologies. The crRNA and tracrRNA powders were diluted to 100 µM with nuclease-free duplex buffer. The 10 µl Hifi Cas9 protein was aliquoted into 10 tubes followed by 1:5 dilution with Opti-MEM (31985062; Thermo Fisher Scientific) solution before use.

Next, we prepared a 10 µM crRNA: tracrRNA duplex solution by mixing 1 µl crRNA stock solution, 1 µl tracrRNA stock solution, and 8 µl nuclease-free duplex buffer and then incubating at 95°C for 3 min in a thermocycler followed by natural cooling at room

temperature for 15 min. Afterwards, we prepared the Cas9/gRNA RNP by mixing 2 $\mu$l Cas9 protein solution and 2 $\mu$l crRNA:tracrRNA duplex solution in 37°C for 10 min. Lastly, we mixed 2 $\mu$l Cas9/gRNA RNP, 5 $\mu$l donor dsDNA, and 0.8 $\mu$l phenol red (P0290; Sigma-Aldrich) and stored it at 4°C. We recommend performing this preassembly step the day before injection.

## Microinjection and sorting for mosaic F0

We injected 1–2 nl Cas9 RNP and donor dsDNA (50–70 pg/nl) into zebrafish embryos at the early one-cell stage. The overall mortality rate was around 50% and we sorted out all dead embryos in the following days. We selected mosaic F0 at 2 dpf based on the fluorescence of the skin (*krt92* and *krt4*), hindbrain and spinal cord (*nkx6.1*), and hindbrain, spinal cord, and olfactory organ (*id2a*) under a wide-field fluorescence microscope LEICA M165 FC (Leica Microsystems) using either the GFP (EGFP or mNeonGreen) or YFP (mNeonGreen) channel. Positive mosaic F0 were put into the fish facility at 6 dpf.

## Genotyping of F1

The clipped zebrafish fins were added to a lysis buffer (10 mM Tris–HCl pH 7.5, 1 mM EDTA, 50 mM KCL, 0.3% TWEEN 20) and boiled at 95°C for 10 min. Next, we added 10% volume proteinase K (10 mg/ml) and incubated it at 55°C overnight. On the following day, we heat-inactivated proteinase K by boiling at 95°C for 10 min and used 1 $\mu$l as the template for PCR reactions. The following primers were used to amplify the fragments over the insertion site, and the reverse primers were used as the sequencing primer:

*krt92* (forward primer): 5′-CAAGCTCAAGCTCAAGTTCC-3′
*krt4* (forward primer): 5′-GTTATGGTGGTAGCGGCTCTGG-3′
*nkx6.1* (forward primer): 5′-CGACGACGACTACAATAAACC-3′
*id2a* (forward primer): 5′-CTCGACTCCAATTCGGCG-3′
EGFP (reverse primer): 5′-CATGTGGTCGGGGTAGCG-3′
mNeonGreen (reverse primer): 5′-ACTGATGGAAGCCATACCCG-3′.

## Quantitative RT–PCR

qRT-PCR was performed using SYBR Green on a ViiA 7 Real-time PCR machine. The gene encoding b-actin was used as the control for normalization. The primer sequences are as follows:

*krt92* (forward primer): 5′-CCGAAACCCTCACCAAGGAA-3′
*krt92* (reverse primer): 5′-CCTCGCTCGTAGATTGGGAG-3′
*krt4* (forward primer): 5′-AACAAGCGTGCTTCCGTAGA-3′
*krt4* (reverse primer): 5′-GCGATCATGCGGTTGAGTTC-3′
*nkx6.1* (forward primer): 5′-CGTGCTCACATCAAAAC-3′
*nkx6.1* (reverse primer): 5′-CGGTTTTGAAACCACACCTT-3′
*id2a* (forward primer): 5′-CAGATCGCGCTCGACTCCAA-3′
*id2a* (reverse primer): 5′-CAGGGGTGTTCTGGATGTCCC-3′
*b-actin* (forward primer): 5′-CGAGCAGGAGATGGGAACC-3′
b-actin (reverse primer): 5′-CAACGGAAACGCTCATTGC-3′

qRT-PCR data are expressed as relative fold change ($\Delta\Delta$Ct). Four zebrafish larvae were pooled as biological replicates. We have four

biological replicates for each knock-in and WT. Two-tailed *t* test was used with *P*-value = 0.05 as statistically significant.

## Lineage tracing by tamoxifen-inducible Cre recombinase

We used both iCre and CreERT2 lines for the testing of Cre and the genetic lineage-tracing experiment. The knock-in iCre lines were crossed with either *Tg(ubi:Switch)* or *Tg(ubi:CSHm)*. For temporal labelling, we treated the zebrafish larvae carrying knock-in CreERT2 and *ubi:CSHm or ubi:Switch* transgenes with 20 $\mu$M 4-OHT (Sigma-Aldrich) in an E3 medium in 24-well plates, four to eight embryos/larvae per well, for 24–36 h without refreshment. Upon induction by 4-OHT, cytoplasmic CreERT2 would be translocated into the nucleus to excise the DNA in between the two loxp sites to enable downstream mCherry or H2BmCherry expression.

## Sample fixation for immunostaining

Before fixing the zebrafish larvae/juveniles, we confirmed the presence of the knock-in cassettes or the sorting markers by examining the corresponding fluorescence signals using a LEICA M165 FC fluorescence microscope. We euthanized the zebrafish juveniles with 250 mg/l tricaine (Sigma-Aldrich) in the E3 medium. Before fixation, we washed the zebrafish larvae/juveniles with distilled water three times. We fixed the samples in 4% formaldehyde (Sigma-Aldrich) in PBS (Thermo Fisher Scientific) at 4°C for at least 24 h. After three washes with PBS, we removed the skin and crystallized yolk (in larvae) by forceps under the microscope to expose the pancreas and liver for immunostaining.

## Hybridization chain reaction

Hybridization chain reaction v3.0 was performed following the protocol described by Choi et al (2018). In brief, the larvae were fixed in 4% PFA at 4°C overnight, with subsequent perforation in 100% methanol at −20°C for at least 24 h. After that, we performed gradient rehydration steps using 75%, 50%, and 25% methanol and five washes using 100% phosphate-buffered saline solution with 0.1% TWEEN 20 (PBST). The larvae were then permeabilized using 30 $\mu$g/ml proteinase K for 45 min at room temperature, followed by 20 min post-fixation using 4% PFA and five washes with PBST. The larvae were incubated overnight at 37°C in hybridization buffer containing 2 pmol of probe set. On the next day, the solution was washed off four times with a washing buffer at 37°C, followed by 5× saline-sodium citrate with 0.1% TWEEN 20 (SSCT) incubation at room temperature for 15 min each. Next, the larvae were incubated in an amplification buffer with 15 pmol of fluorescently labelled hairpin amplifier overnight at room temperature. On the next day, the larvae were incubated with DAPI (1:1,000, in 5× SSCT) for 30 min at room temperature followed by four washes with 5× SSCT. Probe sequences were designed by the manufacturer (Molecular Instruments). The probe set used was: dr_krt4-B1. The conjugated hairpin amplifier is B1, with fluorophore 647.

### Confocal imaging of live zebrafish larvae

We prepared 1% low-melting agarose gel by dissolving 0.01 g agarose (A9414; Sigma-Aldrich) in 1 ml E3 solution with 250 mg/l tricaine followed by 65°C heating for 10 min and cooling on a wedged zebrafish mold. Zebrafish larvae were euthanized in the E3 medium with 250 mg/l tricaine and then repositioned in the gel groove to a suitable position. The confocal imaging was performed using the laser scanning microscopy platform Leica TCS SP8 (Leica Microsystems) with a 10× objective.

### Immunostaining and confocal imaging

We performed immunostaining similarly to our previous report (Liu et al, 2021). In brief, we firstly incubated the zebrafish samples in blocking solution (0.3% Triton X-100, 4% BSA in PBS) at room temperature for at least 1 h. We then incubated the samples in the blocking solution with primary antibodies at 4°C overnight. After removing the primary antibodies, we washed the samples with washing buffer (0.3% Triton X-100 in PBS) 10 times at room temperature for at least 4 h in total. Next, we incubated the samples in the blocking solution with fluorescent dye-conjugated secondary antibodies and the nuclear counterstain DAPI (Thermo Fisher Scientific) at 4°C overnight. Afterwards, we removed the secondary antibodies and DAPI and washed the samples with washing buffer 10 times at room temperature for at least 4 h. The following primary antibodies were used: chicken anti-GFP (1:500, GFP-1020; Aves Labs), goat anti-tdTomato (1:500, MBS448092; MyBioSource), mouse anti-mNeonGreen (1:50, 32F6; ChromoTek), rabbit anti-Insulin (1:100, customised; Cambridge Research Biochemicals), mouse anti-Glucagon (1:50, G2654; Sigma-Aldrich), rabbit anti-Cdh17 (1:1,000; customised sera, gift from Prof. Ying Cao, Tongji University), and rabbit anti-Vasnb (1:1,000; customised sera, gift from Dr. Paolo Panza).

Before confocal imaging, we mounted the stained samples in VECTASHIELD Antifade Mounting Medium (Vector Laboratories) on microscope slides with the pancreas or liver facing the cover slips. We imaged the pancreas and liver with the Leica TCS SP8 platform.

### Hepatocyte ablation by chemo-genetic and pharmacological approaches

The extreme hepatocyte injury model was induced based on the metronidazole/nitroreductase (MTZ/NTR) system (Curado et al, 2008). In brief, the *TgKI(id2a-EGFP-t2a-CreERT2); Tg(ubi:Switch); Tg(fabp10a:CFP-NTR)* larva were treated with 10 mM MTZ (final concentration) in E3 medium (Choi et al, 2014). The severe hepatocyte injury model included incubating the zebrafish larvae in E3 medium supplemented with 10 mM acetaminophen (A7085; Sigma-Aldrich) and 0.5% ethanol for 48 h from 3–5 dpf, followed by three washes with the E3 solution and then 2 d of recovery, as previously reported (North et al, 2010).

### Statistical analysis and data visualization

Similar experiments were performed at least two times independently. The number of cells in the confocal microscopy images was all quantified manually with the aid of the Multipoint Tool from ImageJ. The knock-in strategy scheme was illustrated by using "IBS" software (Liu et al, 2015). Statistical analyses were carried out by two-tailed *Mann–Whitney U* tests (comparing two groups) or by *Kruskal-Wallis* tests (comparing three groups) unless otherwise stated. The results are presented as the mean values ± SEM and *P*-values ≤ 0.05 are considered as statistically significant. The *n* number represents the number of zebrafish in each group for each experiment, and all raw numbers for quantifications can be found in the Table S1. All statistical analyses and data visualizations were performed on R platform (version 4.0.2) using "ggplot2" and "ggpubr" packages.

## Data Availability

The specifics of the key resources are available (Table S2), the construct maps and the Sanger sequencing files that support the correct in-frame integration at the 5′ end of the integrated sequences are available in the public depository (https://osf.io/tdkvh/).

## Supplementary Information

## Acknowledgements

We thank Dr. Ka-Cheuk Liu and Dr. Kyle Mamounis for critical reading of the manuscript; Ying Cao (Tongji University) and Paolo Panza (Max Planck Institute for Heart and Lung Research) for antibodies; and Donghun Shin (University of Pittsburgh), Nikolay Ninov (CRTD Dresden) and Elke Ober (University of Copenhagen) for sharing the fishlines. Some of the illustrations in Fig 1 were created by Biorender.com. Research in the lab of O Andersson was supported by funding from the European Research Council under the Horizon 2020 Research and Innovation programme (grant n° 772365); the Swedish Research Council; the Novo Nordisk Foundation; and the Strategic Research Programme in Diabetes at the Karolinska Institutet. J Mi was supported by the China Scholarship Council (CSC).

### Author Contributions

J Mi: conceptualization, data curation, formal analysis, investigation, visualization, methodology, and writing—original draft, review, and editing.
O Andersson: conceptualization, formal analysis, supervision, funding acquisition, validation, investigation, visualization, project administration, and writing—review and editing.

### Conflict of Interest Statement

The authors declare that they have no conflict of interest.

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
