## [Reviewer comments · Life Science Alliance]

Life Science Alliance

Efficient knock-in method enabling lineage tracing in zebrafish

Jiarui Mi and Olov Andersson

DOI: <https://doi.org/10.26508/lsa.202301944>

Corresponding author(s): Olov Andersson, Karolinska Institutet

Review Timeline:

Submission Date:	2023-01-24
Editorial Decision:	2023-02-20
Revision Received:	2023-02-22
Accepted:	2023-02-23

Transaction Report:

Please note that the manuscript was reviewed at Review Commons and these reports were taken into account in the decision-making process at Life Science Alliance.

Manuscript number: RC-2022-01425

Corresponding author(s): Olov, Andersson

1. General Statements

We really appreciate the reviewers' insightful comments, which help improve the quality of this work. We have responded to the reviewers' questions/comments point by point in the following text and made the corresponding changes in the revised manuscript. Lastly, we added one more figure (Figure 7) with lineage tracing experiments demonstrating the conversion of *id2a*⁺ liver ductal cells to hepatocytes in extreme hepatocyte loss condition.

Reviewer #1 (Evidence, reproducibility and clarity (Required)):

Mi and Andersson describe a method for creating efficient 3' knock-ins in zebrafish using a combination of end-modified dsDNA and Cas9/gRNA RNPs. They tested their method on four genetic loci where they introduced Cre recombinase endogenously, and obtained high F0 mosaicism and germline transmission. The authors included fluorescent proteins with self-cleaving peptides to determine that endogenous expression patterns are observed. By crossing their knock-in Cre lines with lineage tracing reporter lines, the authors temporally traced lineage divergences in zebrafish liver and pancreas.

The authors should clarify the following points before I can recommend publication:

1. Overall, I suggest that the authors consider paring down their figures. Throughout the paper, multiple figure panels convey the same point but for different genes. Furthermore, many construct configurations are shown that are not used in the subsequent panels. For example, the mNeonGreen only (no Cre) constructs and the EGFP constructs are largely not used in downstream experiments. The authors could pick the important constructs and show the relevant data, and summarize all their other constructs in one supplementary figure. The authors also jump around in different parts of the paper with regards to using iCre or CreERT2 and ubi:Switch or ubi:CSHm. It's not clear to me why they're doing that? It makes the paper hard to follow. For example, why use iCre - it's not temporal if I understand correctly (and I'm not sure what improved Cre is - could they reference a paper and include a small explanation) so CreERT2 seems suitable especially for their temporal lineage tracing experiments. Why not limit the description to CreERT2 in the main text/figures? Also, isn't ubi:Switch and ubi:CSHm pretty similar except the latter is nuclear mCherry due to H2B? Why not only focus on ubi:CSHm experiments? I found the paper to be unnecessarily long and think it would benefit from editing to describe the most important concepts and experiments.

Thank you for your constructive and helpful comments. We do agree that sometimes the schematic constructs seem redundant. This is because the *krt4*, *nkx6.1*, and *id2a* genes have similar gRNA targeting sites (all spanning over the stop codon). However, we prefer to keep these schematic constructs as we have all the statistical results showing the knock-in efficiency in the subsequent figure panels. Such layout can allow readers to make comparisons and better understand the efficacy of this method. However, combined with the comments from the second reviewer, we indeed need to add more detailed information, including the sequence and the length of the short left and right homologous arms in the schematics, to enable the readers to follow this strategy more easily. Meanwhile, we added a new supplementary figure with the sequences of the long left and right homologous arms, as well as the genetic cassettes/point mutations for *krt92* knock-in (Figure S1).

As for the color switch lines we used, we appreciate your comments and replaced Figure 5E-G with new fluorescent images using zebrafish larvae carrying the *ubb:CSHm* transgene. For most of the lineage tracing experiments in this study, we used *Tg(ubb:CSHm)* as the H2BmCherry is more stable, located in the nucleus, and the fluorescence intensity is stronger than in *Tg(ubb:Switch)*. However, for the lineage tracing experiments of the liver injury model, we believe that *Tg(ubb:switch)* is a better option than *Tg(ubb:CSHm)*. In the absence of a hepatocyte specific far-red reporter line, we can distinguish the hepatocytes derived from the *id2a*⁺ origin using the *Tg(ubb:Switch)* line, as the cells with Cre recombination express mCherry in the cytoplasm; i.e. we can tell the cell types based on the cell morphology in combination with the ductal anti-vasnb staining. This strategy was previously used by Dr. Donghun Shin's group in their 2014 Gastroenterology paper (Figure 4B, DOI: 10.1053/j.gastro.2013.10.019). Therefore, we still kept the *ubb:switch* in the Figure 1F schematic, and we have elaborated on why we chose *Tg(ubb:switch)* line for the *id2a*⁺ cell conversion experiments in Figure 7 and S14.

The iCre we used is a codon-improved Cre (iCre). The original cDNA sequence was from pDIRE (Addgene plasmid #26745; provided by Dr. Rolf Zeller, University of Basel) (Osterwalder et al., 2010).

At the beginning of this project, we actually didn't know whether there were any differences between iCre and CreERT2 in labelling of the cells of interest. Here, using both the iCre and CreERT2 lines, we for the first time, formally show the developmental lineage path of *nkx6.1*-expressing cells in the zebrafish pancreas. Our data suggested that the early *nkx6.1*-expressing cells are multipotent pancreatic progenitors giving rise to all three major cell types in the pancreas (endocrine, ductal and acinar cells, shown by *nkx6.1* knock-in iCre) and gradually the *nkx6.1*-expressing cells become restricted in the ductal/endocrine lineages (shown by the *nkx6.1* knock-in CreERT2 treated with 4-OHT at different timepoints). In addition, we also aim to use these knock-in lines for multiple studies in which we need to perform many quantitative experiments. As expected, we are unable to reach 100% labeling using the knock-in CreERT2 lines, even if we treated the larvae with very high concentration of 4-OHT over a long period of time. This means that the CreERT2 induced recombination will introduce more variation for quantitative experiments (for instance, the number of regenerated beta-cells from the ductal origin). As we were quite confident with the efficiency of this knock-in strategy, we decided to make both iCre and CreERT2 lines in *krt4*, *nkx6.1*, and *id2a* locus and just observe how they performed. We often

use iCre knock-in lines for lineage tracing experiments, because the iCre lines reach near 100% labeling efficiency. Such iCre lines are particularly useful if they only label terminally differentiated cell types. Thus, the near 100% labeling efficiency in iCre lines can be of great help for initial experiments, which later can be confirmed by temporal labeling using CreERT2 lines.

2. Could the authors describe the purpose of the 5'AmC6 modification earlier in the paper? I didn't see much text about it until the discussion. It seems that the speculation is that it provides end protection and prevents degradation (based on *in vitro* studies in human). This should be inserted into the introduction as a reader might be wondering about this and won't find an answer until near the end. Also, is this the first *in vivo* use of this modification for knock-ins? If so, that should be highlighted in the text.

This is a helpful comment. In the revised manuscript, we elaborate more on why we chose 5'AmC6 modification in our donors. To our knowledge, this is the first time this 5' modification is used *in vivo*, however, bulky 5' modification (5'Biotin - 5x phosphorothioate bonds) has been used in medaka (DOI: <https://doi.org/10.7554/eLife.39468.001>, 2018 Elife, as we previously referenced). The cell division rate is much faster in zebrafish embryos compared with medaka embryos during early development, so we speculate that such modification might be of more importance in zebrafish to achieve early integration. Another advantage is that the 5'AmC6 modification is commercially available, allowing researchers to prepare the donor dsDNA in a handy fashion. We have now expanded on these details and advantages in the introduction.

We have now also performed a head-to-head comparison between AmC6- and Biotin-end protection for knock-in at the *krt4* locus. To our surprise, we noticed a much higher knock-in efficiency using AmC6 (35/428 embryos had >30% skin with green fluorescence label) compared to biotin (3/526 with only visible green). Such dramatic difference suggests the AmC6 modification, at least at this locus, is a better option than biotin. However, we are reluctant to make a generalized conclusion that AmC6 is better than biotin, since we are not able to test it accurately on most genes since the expression level isn't high enough to assess integration in individual cells (contrary to methods using eye- or heart-markers).

3. The authors do not show any sequencing data confirming that their insert was knocked-in as designed with no disruption to the immediate upstream and downstream endogenous sequences. Can they sequence the loci to confirm?

This is indeed a question we frequently get – thank you for making us relay this information more clearly! We have put the raw Sanger sequencing data in a public repository (mentioned in the Data Availability section), and included the sequencing primers in the method paragraph. Now we also refer to this data in the discussion section in conjunction to highlighting that the integrations were correctly placed in the loci. If you think there are better ways to show the sequencing results, please let us know.

4. I found the descriptions of the long and short HA to be confusing when describing the results, especially since the first tested gene *krt92* only has long and all subsequent ones are short. The discussion made it more clear that short HA is more efficient and applicable when gRNAs span the stop codon. Perhaps that wasn't possible with *krt92*, but the authors could prevent the confusion by clearly stating the design requirements of long and short HA and that they wanted to test which is more efficient before starting to describe the data. I also didn't see a description of what the length difference between long and short HA is? How short is short HA?

This is a great question that is well worth discussing. In the revised manuscript, we changed the order in which the parts are described, with *nkx6.1* knock-in in front of *krt4* knock-in. Here we explain why we would like to do that:

At the beginning of this project, we did not know if the 5' modified dsDNA could be an effective donor. To test our hypothesis, we chose the *krt92* gene as our first target, as this is a keratin protein and expressed in the epithelial cells. We can easily detect the fluorescence in the epithelial cells (most notably in the skin), which allow us to sort the F0 mosaic embryos with high percentage of integration. Notably, from our experience, the most difficult part of the knock-in method is the sorting step (usually performed during 1-3 dpf). This is because the fluorescence signal is highly dependent on the endogenous gene expression level and is usually dimmer with an overall integration efficiency that is lower compared to canonical transgenesis. Therefore, we thought that targeting an epithelial cell marker would be informative and help us to evaluate the validity and reproducibility of the method. If it worked, then we could move on targeting genes expressing in more restricted tissues or cell types. For *krt92* gene, the gRNA targets the region upstream of the stop codon. To prevent the cleavage of the donor template, we had to introduce several point mutations and at the same time keep the amino acid sequence intact. However, such mutations can restrict the knock-in and lower the integration efficiency when using shorter arms (due to the sequence mismatch).

After we managed to make the *krt92* knock-in, our next question was, what about using a gRNA spanning over the stop codon region? In this way, we don't need to introduce point mutations on neither the left nor the right homologous arm. Also, for the purpose of our biological study, the *nkx6.1* were on top of our gene list for lineage tracing experiments and we luckily identified that there is very good gRNA targeting this locus. After we successfully made the *nkx6.1* knock-in, we were thinking that we could simplify the protocol even further, i.e. switching to short homologous arms so that we can prepare the donor by a one-step PCR instead of making complicated constructs. We tested that hypothesis in *nkx6.1*, *krt4*, and *id2a* sites and obtained very promising efficiency. Also, we did some further testing with dsDNA without the 5' modifications and showed that the 5' modifications indeed greatly increased integration efficiency. Therefore, although the short homologous arm method is a highlight here, we also point out that it was not planned from the beginning. In the revised manuscripts, we want to convey our method in a logical way and show how we modify the method in a step-by-step fashion.

Moreover, with regards to the comments from the second reviewer, we now added the length of the homologous arms as well as the mutation site on the schematics. We chose short

homologous arm because in previous literature it was suggested that short homologous arms (36-48 bp, which we now write out in both the results and the methods) can promote microhomology-mediated end joining (doi: 10.1096/fj.201800077RR). We also noticed that the recent Geneweld method (DOI: 10.7554/eLife.53968) also adheres to a similar length for homology mediated integration. In this study, HAs even shorter than 36 bp also perform well.

5. The authors state that they could not use *in situ* to confirm *krt92* endogenous and knock-in expression overlap, but rather say that they match based on data from an intestine scRNA-seq dataset. Can they elaborate on this? Which clusters/cell types show overlap? Furthermore, is there any *krt92*:GFP transgenic line that can be used as a reference for expression as well? This point is also applicable for *krt4* described in Fig.2

We appreciated this point. In the beginning, we contacted Molecular Instruments to synthesize *krt92* HCR3.0 *in situ* hybridization probes. However, the technical staff there told us that they are unable to make specific probes due to high sequence similarity to other keratin protein families. We can see that the sequence similarity mostly occurs in the middle of *krt92* genes, and the HCR3.0 probes rely on a probe set (preferably 20-30 probes with different sequences) to target the mRNA.

The scRNA-seq data that we referenced are from 10X platform, which is based on a 3' enrichment methodology. The reads mapping to *krt92* genes are mostly located on the 3' end. This is good as there is much less similarity to other cytoskeleton genes in the 3' end of the gene. Unfortunately, there is no *krt92* transgenic lines available, so we relied on the single-cell data to correlate expression patterns in this case.

There are two zebrafish intestine single-cell data sets available, with the following links:
(1): https://singlecell.broadinstitute.org/single_cell/study/SCP1675/zebrafish-intestinal-epithelial-cells-wt-and-fxr?genes=krt92#study-visualize
(2): https://singlecell.broadinstitute.org/single_cell/study/SCP1623/zebrafish-intestine-conventional-and-germ-free-conditions?genes=krt92#study-visualize

We can see that *krt92* is widely expressed in different types of intestinal epithelial cells (absorptive enterocytes, secretory enteroendocrine/goblet cells and ionocyte).

[Figure removed by LSA Editorial Staff per authors' request]

For the *krt4* gene, we now added the HCR3.0 *in situ* hybridization and immunofluorescence for both *krt4* knock-in EGFP-t2a-CreERT2 lines and the *Tg(krt4:EGFP-rpl10a)* transgenic line (a construct from Anna Huttenlocher, <https://www.addgene.org/128839/>, which has been widely used to label skin cells). The results are shown in Figure S9. We show that *krt4* has very high expression in the intestinal bulb and hindgut based on the HCR3.0 *in situ*. The Immunofluorescence of the *krt4* knock-in fully recapitulate the *krt4* expression pattern in the intestine, while there is almost no fluorescence signal in *Tg(krt4:EGFP-Mmu.Rpl10a)*. We believe this is another advantage of using the knock-in method, over transgenics, for cellular labeling and lineage tracing. Classical transgenics often rely on short promoters of the proximal/enhancer region upstream of ATG with various length (arbitrarily or based on clues from motif analysis/DNA methylation sites). However, different tissues/cell types tend to use different *cis*-regulatory elements and the chromatin structure/enhancer-promoter loops might differ dramatically among different cell types. It is hard to predict the exact region of the regulatory sequences that is

sufficient for driving the gene expression in a certain cell type. Thus, such reasoning consolidates with that our knock-in lines recapitulate the endogenous *krt4* gene expression. Therefore, we believe that the knock-in based genetic lineage tracing will become the standard in the zebrafish field, as theoretically it avoids both the lack of relevant expression and leakage problems of transgenics.

6. I think Figure 2A needs the dotted lines on the last construct to be fixed (points to p2A)

Thank you for noticing! This was due to a bug in the IBS software, and we changed it manually using Adobe Illustrator in the revised manuscript.

7. There are a few instances where the authors describe performing 4-OHT treatment for long period (e.g. over a 20 hour or 24 hour period). Is fresh 4-OHT added after a certain amount of time or is it a one-time addition? Is such long periods of 4-OHT required or has maximal recombination already occurred within a few hours after addition of 4-OHT?

For 4-OHT treatment, we referred to the method described by Dr. Christian Mosimann (DOI: 10.1371/journal.pone.0152989). We actually tried different conditions (dosage, duration, refresh or not). This is particularly important for the knock-in CreERT2 lines because the level of CreERT2 is highly dependent upon the endogenous gene expression level. In our case, the *nkx6.1* and *id2a* are transcriptional regulators and relatively lowly expressed compared with structural proteins. We maximized the labeling efficiency by using the highest concentration and longest duration suggested for 4-OHT treatment. The 4-OHT was stored in -20 °C and it would become less effective after 30 days of storage. Therefore, we first incubated the 4-OHT in 65 °C for 10 min (as recommended by Dr. Christian Mosimann) in order to convert it to a bioactive form. Next, we treated the zebrafish embryos with 4-OHT using a final concentration of 20 μM for 24 hours. We didn't refresh the 4-OHT since there was no significant difference compared with a one-time addition. Moreover, using higher dosage or longer treatment time can lead to less survival and increased deformity rate. 20 μM 4-OHT treatment for shorter time periods (6 or 12 hours) can cause high labeling variability (some larvae have good labeling while others not). In the end, after several rounds of experiments, we settled on 20 μM 4-OHT treatment for 24 hours as it can reach the highest labeling efficiency, lower variability, and good survival.

8. For Figures 4-6 where confocal images of lineage tracing experiments are shown, there is no indication of how many times the experiments were repeated, how many sections were images, how many animals used, how many cells counted. All of this information should be included in the figure legends and plots should be added showing quantification and statistical analysis (where appropriate).

The reviewer makes a good point and we have now added the number of larvae used and statistical results for the quantitative experiments. The quantification of experiments in Figure 3E-H (originally Figure 4E-H) are shown in Figure S6D using box/dotplot. We randomly selected 3 secondary islets of different sizes (large, middle, and small) from each juvenile fish (n=5) and pooled the number of mCherry/ins double positive cells and ins positive cells together. The quantification of the lineage-tracing efficiency in the experiments in Figure 6 are shown in Figure S13.

9. Figure 4 C, C' - I'm not sure what to look for. Is the message that there is no Cherry positive cells that are *vasnb* negative when labelling is done at 8 somite? But the *vasnb* positive cells that are also Cherry positive remain? The *vasnb* staining seems much weaker/harder to see in C C' compared to B, B'. As mentioned above, these data should be quantified and statistical significance indicated.

Thank you for pointing this out; the second reviewer made a similar point. We redid the experiments using zebrafish larvae carrying the *ptf1a:EGFP* transgene to indicate the acinar cells (Figure 3B-D, Figure S4G). We also quantified the results and performed statistical testing.

10. I recommend the authors include a short section in the discussion comparing the efficiency of their method to other knock-in strategies used in zebrafish. This is an important claim of the paper yet it is not clear how much better it is (if at all) in terms of frequency of F0 mosaicism and identification of founders relative to other methods. I do appreciate the relative simplicity of the molecular steps of construct design/generation.

This is indeed important. It is also tricky since we are unable to make head-to-head comparisons between different methods as we are targeting different genetic loci and do not have the other methods up and running in our lab. However, the general comparison is based on the statistics shown in the hallmark papers describing these other methods, regardless of which genes were selected for targeting. In the discussion, we added a list of points that are novel/improved with our method versus previous ones, including that: 1) we simplify the knock-in methodology circumventing complicated molecular cloning; 2) we have very high germline transmission rate, which means that one morning of injection is often enough to get a founder; and the expression of fluorescence proteins avoids tedious work in identifying founders, which also saves a lot of space in the fish facility; 3) our lines can be applied for multiple utilities; 4) the method does not disrupt the endogenous gene product. We believe this is critical for the field of developmental biology, regenerative medicine, and disease modeling in zebrafish – and perhaps a similar 3' knock-in based lineage-tracing method can become commonly used to delineate the cell differentiation and plasticity during homeostatic and diseased conditions in additional organisms.

Full Revision

Reviewer #1 (Significance (Required)):

Overall, the study contributes a new knock-in strategy in zebrafish that appears to be more user-friendly and results in high germline transmission. The authors also identify nkx6.1+ ductal cells as progenitors of endocrine cells in the pancreas highlighting the biological applications of their method. I think this study represents an important advancement in zebrafish genetics and will have future impact in lineage tracing during development, regeneration, and disease.

Reviewer #2 (Evidence, reproducibility and clarity (Required)):

Summary:

Here, the authors present a strategy where they performed knock-in at the level of the STOP codon, taking care of not perturbing the coding region. They integrate cassettes coding for fluorescence protein and Cre recombinase, which are separated from the endogenous gene and each other by two self-cleavable peptides.

The cassettes are done by PCR with primers with 5' AmC6 modifications and they test short (36 to 46 bp) or long homologous arms (~950bp). For nkx6.1 gene, they observed a dramatic increase of recombination efficiency when injecting the donors with short Homology arms compared to long arms suggesting that short arms could be used. Indeed, short arms used with krt4 and id2a allow them to obtain K.I lines.

The techniques described here look promising. Indeed, even if the proportion of F0 showing adequate reporter expression is low (usually about 2%), the percentages of founders among these mosaic F0 were quite high (between 50% and 100%). And this is the most important aspect as it is usually the most time-consuming aspect of the work.

Major comment:

The authors claim that the knock-in lines can precisely reflect the endogenous gene expression, as visualized by optional fluorescent proteins. But are the authors sure that the integration of the cassettes coding for fluorescence protein and Cre recombinase, which are separated from the endogenous gene and each other by two self-cleavable peptides, will not affect the level of expression of the targeted genes. Indeed, it has been shown that sometimes self-cleavable peptides could affect the expression of the genes of the cassette like for example in this reference (<https://www.ncbi.nlm.nih.gov/pmc/articles/PMC8034980/>). Therefore it is important that the authors check whether the cassette affect the level of expression of the targeted gene if they want to claim that the knock-in lines precisely reflect the endogenous gene expression.

Thank you for your insightful comments. With regards to the endogenous gene expression, we now use qPCR for further validation. We added the qPCR results to the supplementary material (Figure S15) in the revised manuscript. In brief, we pooled 4 larvae in one tube per biological

Full Revision

replicate and have 4 biological replicates for each knock-in line. We didn't see a significant change in the endogenous expression for any gene. In addition, we have grown up homozygous knock-in lines to adulthood and they are fertile without any overt phenotype.

The highlighted reference is dealing with a cardiomyocyte specific transgenic line, and we assume figure 3-Supplementary figure 1 is what the reviewer is referring to. The altered level of *erbb2* expression might be due to the experimental conditions (no treatment or 3 days post treatment). Also, it is possible multiple transgenic insertions occur, as well as gene silencing at some insertion sites. However, such issues would not present, or very limited, with knock-in methods.

Minor comments:

General points:

I believed that the authors should improve the presentation of their data. Indeed, based on what they present, it would be impossible for me to reproduce their technique. Indeed, it is not clear at all how they design the short and long arm, where they are exactly located, which mutations they have done (for fig1), where is located the guide RNA compared to the STOP codon and the HA arms. Graphics that exactly place all these sequences are absolutely required to understand the strategy used and should be placed in figure 1, 2, 3 and 4.

Thank you for these comments. In the revised version, we added the sequence information of the short homologous arms in each of the schematics. As for the *krt92* gene, we added the sequence information in the first supplementary results (Figure S1) with the genetic cassettes and point mutation information. We list all the primer information in the methods. Also, we have uploaded our vector templates in the public repository (as listed in the Data availability section). Lastly, we added a key resource table in the supplementary file with all the detailed information of reagents for the ease of reproducibility (including all the primers sequences used). We are also willing to share our constructs with the scientific community upon request.

Specific points:

Introduction:

"In zebrafish, the NHEJ-mediated methods have been intensively investigated in 5' knock-in upstream of ATG using donor plasmid containing in vivo linearization site flanking the insertion sequences (11,12,17-20). The 3' knock-in method has also been examined using circular plasmid as the donor with either long or short homologous arms (HAs) flanked by in vivo linearization sites (14, 21-23). Recently, intron-based and exon-based knock-in approaches have remarkably expanded the knock-in toolbox by targeting genetic loci beyond the 5' or 3' end (8-10,13,24-26)." This part should be explained better in order that the readers could really understand the differences between these old studies and this new one. And really insist on what is the novelty of their technique.

Full Revision

Good points. In the revised version, we elaborated more on the previous discoveries, the major challenges, the knowledge gap in zebrafish knock-in methodology, and what is novel and improved with our new technique. Please, see clarifications and the expanded text in both the introduction and discussion.

Results:

Page 4: To my opinion, the first paragraph should be removed and the technique directly explained based on *krt92* strategy as this paragraph does not allow to understand the technique. As indicated above, figure 1 should indicate more clearly the location of the long arms and which mutations they have done and where is located the guide RNA.

Figure 1G: The expression in the skin is far from obvious and the image should be improved (for example with some inset).

Thank you for the comments. We added a new supplementary figure (Figure S1) and show the sequences of left and right homologous arms, the genetic cassettes, as well as the point mutations with different background color highlight. We added the insets to show the magnified regions of interest. Also, we added the images from the fluorescent microscope used for sorting, to show the EGFP signals in live zebrafish embryos (Figure S2D and Figure S8D).

Figure 3E: The authors say that "cells expressing *nkx6.1* (displayed by the green fluorescence) were located on the ventral side of the spinal cord whereas H2BmCherry positive cells, which include all the progenies of *nkx6.1*+ cells after the iCre recombination, resided in both the ventral and dorsal parts of spinal cord". This differential expression in the spinal cord is not obvious and a more closer view should be provided.

Thank you for the comment. First, we changed the order and now describe all *nkx6.1* content in Figure 2 and 3 and the *krt4* content in Figure 4. We added insets to show the magnified regions and better display the expression pattern of the two fluorescence proteins in Figure 2E-G. One can now clearly see from the magnified insets that the green signals driven by the endogenous *nkx6.1* gene are present in the ventral part of the spinal cord, while the red signals are present in both the ventral and the dorsal side of the spinal cord.

Fig S4H: The authors say that" using lineage tracing, we could trace back all three major cell types in the pancreas (acinar, ductal and endocrine cells) to *nkx6.1* lineage (Figure 3H-H', Supplementary Figure S4G, H)". While this is obvious for endocrine, the colocalisation with *ela3l*:GFP is not obvious and the figure should be improved.

Full Revision

This is a very good point, and the first reviewer gave similar suggestions. In the revised version (shown in Figure S4H and I), we added the insets to show the magnified regions to better display the expression pattern of two fluorescence proteins. The *ela3l* reporter line is using a short promoter to drive the expression of H2B-EGFP (doi: 10.1242/dmm.026633). However, this short promoter cannot reach 100% labeling of acinar cells, so we also use the *ptf1α:EGFP* transgene for further validation (new Figure S4G). Both transgenic reporter lines showed many EGFP and mCherry double-positive cells, indicating that these acinar cells are derived from a *nkx6.1*-expressing origin. Here we did not use the anti-GFP antibody, as our color switch lines contains CFP and anti-GFP antibody can also recognize CFP. However, the GFP signal is strong enough to show the expression. We hope the additional experiments and insets clarifies this point.

Page 8: the authors say that "The immunostaining at 6 dpf showed that both intrapancreatic ductal cells and a portion of acinar cells can be lineage traced when the 4-OHT treatment started at the 6 somite stage (Figure 4B and B'). The identification of the acinar cells has been done based on the absence of the ductal marker *vasnb*. To trace efficiently the acinar cells, this should be done with an acinar marker.

Another good point also mentioned by reviewer one. We redid the analyses using zebrafish larvae containing the *ptf1α:EGFP* transgene to indicate the acinar cells and the co-expression pattern with the lineage-tracing (the data is shown in new Figure 3B-D).

Reviewer #2 (Significance (Required)):

I do not have enough expertise in the KI field to evaluate whether this strategy is really novel and as mentioned above, the authors should better explain what is really the novelty of their strategy.

In our answers to the comments of the first reviewer, we elaborated more on the points that are novel/improved with our method vs previous methods, as reiterated here:

"...including that: 1) we simplify the knock-in methodology circumventing complicated molecular cloning; 2) we have very high germline transmission rate, which means that one morning of injection is often enough to get a founder; and the expression of fluorescence proteins avoids tedious work in identifying founders, which also saves a lot of space in the fish facility; 3) our lines can be applied for multiple utilities; 4) the method does not disrupt the endogenous gene product."

Moreover, the first reviewer asked about the difference between the *krt4* knock-in and *krt4* transgenics, and based on the *in situ* data, we showed that our *krt4* knock-in can fully recapitulate the endogenous gene expression, while the *krt4* transgenics can hardly label the intestinal bulb and hindgut. This might be due to that different tissues/cell types may depend on different *cis*-regulatory elements to drive the gene expression. The chromatin structure and the enhancer/promoter loop might also differ dramatically among different tissues. Therefore, the transgenics might be useful for one type of cells, while they might be not useful at all for other cell types. In the future, we believe that, similar to the mouse field, the 3' knock-in based lineage

Full Revision

tracing methods might become the standard method in the zebrafish field, to delineate cellular differentiation and plasticity during homeostatic and diseased conditions.

February 20, 2023

RE: Life Science Alliance Manuscript #LSA-2023-01944

Dr. Olov Andersson
Karolinska Institutet
Cell and Molecular Biology
von Eulers väg 3
Stockholm, - 171 77
Sweden

Dear Dr. Andersson,

Thank you for submitting your revised manuscript entitled "Efficient knock-in method enabling lineage tracing in zebrafish". We would be happy to publish your paper in Life Science Alliance pending final revisions necessary to meet our formatting guidelines.

- please address the final Reviewer 1's point
- please upload your manuscript text as an editable doc file
- please upload your main and supplementary figures as single files
- please add a Running Title, summary blurb/alternate abstract, and a category to our system
- please add the Twitter handle of your host institute/organization as well as your own or/and one of the authors in our system
- please add the author contributions to the main manuscript text
- please use the [10 author names, et al.] format in your references (i.e. limit the author names to the first 10)
- please add a figure callout for Figure S10J and Figure S12A-D to your main manuscript text

Figure Check:

- please make your scale bars more visible

A. FINAL FILES:

B. MANUSCRIPT ORGANIZATION AND FORMATTING:

Sincerely,

Reviewer #1 (Comments to the Authors (Required)):

The authors have addressed my concerns from the first round, and the paper is acceptable for publication.

My one criticism remains that the figures of the constructs are still a bit confusing since the authors introduce many versions all at the same time but don't refer to them in the same order. For example, the `ubb:loxP-EGFP-loxP-mCherry` line isn't referenced until Fig 7. This might be a personal stylistic comment, and I leave it to the authors and editorial team to decide on the best presentation. As a reader, I did find it took more effort to follow despite having read this paper in the last round. Just my two cents.

February 23, 2023

RE: Life Science Alliance Manuscript #LSA-2023-01944R

Dr. Olov Andersson
Karolinska Institutet
Cell and Molecular Biology
Solnavägen 9
Stockholm, - 171 65
Sweden

Dear Dr. Andersson,

Thank you for submitting your Methods entitled "Efficient knock-in method enabling lineage tracing in zebrafish". It is a pleasure to let you know that your manuscript is now accepted for publication in Life Science Alliance. Congratulations on this interesting work.

DISTRIBUTION OF MATERIALS:

Again, congratulations on a very nice paper. I hope you found the review process to be constructive and are pleased with how the manuscript was handled editorially. We look forward to future exciting submissions from your lab.

Sincerely,
